# Benchmarking and Improving Fine-Grained Text-to-Image Alignment via Paired Reinforcement Learning

**Kaihang Pan** [* 1] **Wendong Bu** [* 1] **Yuruo Wu** [* 1] **Kai Shen** [1] **Yang Wu** [2] **Yun Zhu** [3] **Zehan Wang** [1] **Yunfei Li** [2] **Hang Zhao** [2] **Juncheng Li** [1] **Siliang Tang** [1]

## Abstract

While recent autoregressive models have achieved text-to-image generation performance comparable to diffusion models, they significantly struggle with fine-grained semantic alignment. To rigorously evaluate this limitation, we introduce DeltaBench, a benchmark featuring paired prompts with subtle fine-grained differences, which reveals that existing models fail to achieve precise control over visual tokens. To bridge this gap, we propose FocusDiff, a comprehensive framework that enhances alignment by learning from subtle differences in similar text-image pairs. Specifically, we construct `FocusDiff-Data`, a large-scale dataset of paired samples derived from image editing tasks to capture localized semantic shifts. Furthermore, we introduce Pair-GRPO, an improved reinforcement learning algorithm that extends GRPO to paired samples. Extensive experiments demonstrate that our approach outperforms most prior prominent methods on both DeltaBench and existing benchmarks.

## 1. Introduction

Witnessing the scalability of autoregression (AR) in Large Language Models (OpenAI, 2023), recent studies (Google, 2025b) have introduced the AR paradigm to text-to-image generation, achieving performance comparable to diffusion models (Labs, 2024). Leading AR-based approaches (Geng et al., 2025; Team, 2026b) typically encode images into discrete tokens, thus formulating generation as a next-token prediction task. Then a diffusion decoder is employed to decode the predicted tokens back into the pixel space, effectively harnessing the strengths of both paradigms.

Despite extensive vision-language alignment, existing AR-based models still struggle with precise control over images based on text conditions. To elucidate this problem, we first introduce **DeltaBench benchmark** to evaluate fine-grained semantic adherence. Unlike typical text-to-image benchmarks (Ghosh et al., 2023; Hu et al., 2024) with a single prompt per test case, each test case in DeltaBench comprises a pair of prompts. The paired prompts share a high degree of semantic overlap but contain subtle, word-level distinctions—nuances that models frequently neglect in favor of dominant semantic features. For every prompt pair, we generate corresponding images and evaluate their text-image consistency scores, denoted as $s^1$ and $s^2$. We then calculate the arithmetic mean score as $s_a = (s^1 + s^2)/2$, and the geometric mean score as $s_g = \sqrt{s^1 * s^2}$.

Ideally, models should precisely distinguish the semantic nuances between prompts and accurately generate the corresponding images. However, even for the SOTA AR-related model (Geng et al., 2025; Team, 2026b), the geometric mean in DeltaBench is significantly lower than the arithmetic mean, as shown in Figure 1(a). Considering that the geometric mean is highly sensitive to lower values, the results indicate the instability control over fine-grained visual generation. The examples in Figure 1(b) further illustrate that models often fail to capture details outside the primary semantic focus. We argue that this problem lies in the lack of fine-grained text-to-image alignment. Standard training lacks fine-grained annotations that explicitly map sentence constituents to specific image regions. Thus, the optimization disproportionately prioritizes dominant semantics while neglecting intricate, fine-grained details.

Thus, a crucial question emerges: ***How can we achieve robust fine-grained text-image alignment to enable precise control over visual semantics in AR-based text-to-image generation?*** While recent advances (Yin et al., 2024; Zhao et al., 2024) in image comprehension have attempted to bridge this gap by leveraging contrastive learning to enforce intra-sequence token alignment, they undermine the core design philosophy of AR, failing to fully leverage the successful infrastructure of LLMs. We aim to find an elegant solution for fine-grained text-image alignment without

---

[*]Equal contribution  [1]Zhejiang University [2]Ant Group [3]Shanghai Artificial Intelligence Laboratory. Correspondence to: Siliang Tang <siliang@zju.edu.cn>.

*Proceedings of the $43^{rd}$ International Conference on Machine Learning*, Seoul, South Korea. PMLR 306, 2026. Copyright 2026 by the author(s).

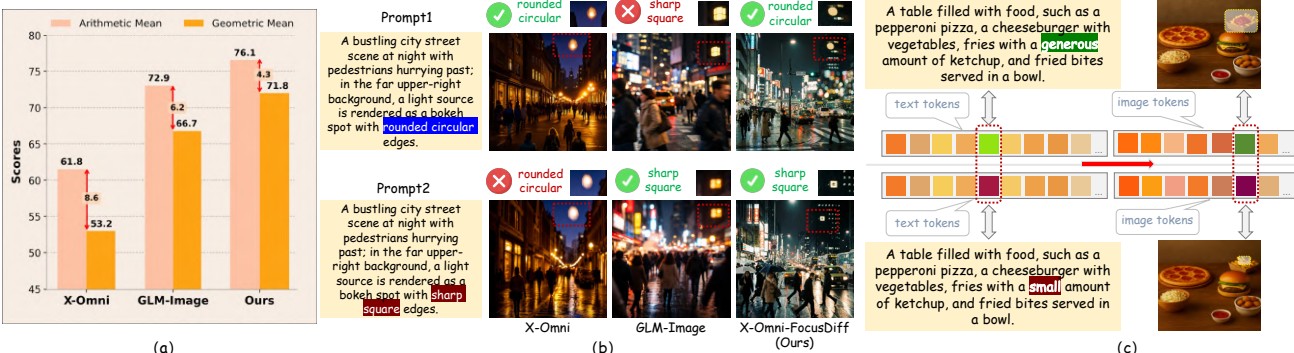

*Figure 1.* (a) Geometric/arithmetic mean score in DeltaBench for AR-based models. (b) Examples of generating images for similar prompts. (c) The subtle sensory differences between images or between texts result in only minor alterations to specific tokens.

altering the original AR-based training paradigm.

In this paper, we introduce **FocusDiff**, a method that enhances fine-grained text-image semantic alignment by learning from the subtle differences between similar text-image pairs, without disrupting the original AR-based training paradigm. Specifically, **from the data perspective**, we introduce **FocusDiff-Data**, expanding the training case from a single text-image pair $\{(\mathcal{T}, \mathcal{I})\}$ into a set of two pairs $\{(\mathcal{T}^1, \mathcal{I}^1, \mathcal{T}^2, \mathcal{I}^2)\}$. Here, $\mathcal{T}^1$ and $\mathcal{T}^2$, as well as $\mathcal{I}^1$ and $\mathcal{I}^2$, appear similar in overall expression, but differ in fine-grained details beyond the core semantics. For example, $\mathcal{T}^1$ is consistent with $\mathcal{I}^1$ but not with $\mathcal{I}^2$, and vice versa. As shown in Figure 1(c), the subtle sensory differences between images or between texts result in only minor alterations to specific visual or textual tokens. By leveraging these differential signals, FocusDiff enables MLLMs to trace the causal efficacy of textual variations on specific visual token trajectories, thus establishing fine-grained associations between the two modalities.

**From the training perspective**, we introduce Pair-GRPO, a paired reinforcement learning method that guides the model in learning fine-grained semantic differences through an exploration-exploitation trade-off. Specifically, we extend the Group Relative Policy Optimization (GRPO) (Shao et al., 2024) to visual generation with two key improvements:

**(1) Expanding the Group Concept:** While vanilla GRPO considers $G$ responses from the same prompt as a group, we expand this to include $2 \times G$ responses from pairs of similar prompts with fine-grained semantic differences from FocusDiff-Data.

**(2) Shifting Focus from Exploitation to Exploration:** Unlike vanilla GRPO, which encourages fully autonomous exploration without ground-truth images, we provide ground-truth images from FocusDiff-Data during early training to enhance exploration and guide the model to better grasp fine-grained semantic differences. As training progresses, we gradually anneal this supervision, transitioning

from exploitation-first to exploration-first.

On this basis, with X-Omni (Geng et al., 2025) as the backbone, we realize better fine-grained text-image semantic alignments on both DeltaBench and existing benchmarks. Our main contributions are threefold:

- We introduce DeltaBench benchmark, featuring test cases with two prompts only differing in fine-grained, easily overlooked semantics, highlighting existing models' limitations in precise visual control.

- We propose FocusDiff, a paired text-image dataset with an improved GRPO-based training paradigm, focusing on fine-grained semantic differences to boost text-image alignment.

- We achieve superior performance on existing text-to-image benchmarks and significantly outperform most prior prominent methods on DeltaBench.

## 2. Benchmark: DeltaBench

**Data format and Task Categorization.** In traditional text-to-image benchmarks (Ghosh et al., 2023; Hu et al., 2024), each test case consists of a single prompt, which is used to measure the overall semantic alignment between the prompt and the generated image. In this section, we introduce a new benchmark called **DeltaBench**. Each test case in DeltaBench contains two similar prompts with subtle differences that are often overlooked beyond the primary semantics. By comparing the accuracy of the images generated by the model for each prompt, we evaluate whether the model has focused on the fine-grained semantic differences in the prompts to generate correct images. We design six categories of word-level subtle differences: (1) Overall appearance; (2) Color; (3) Counting; (4) Position; (5) Style & Tone; (6) Text. And We present examples of each category in Figure 2(a). See more details in Appendix C.

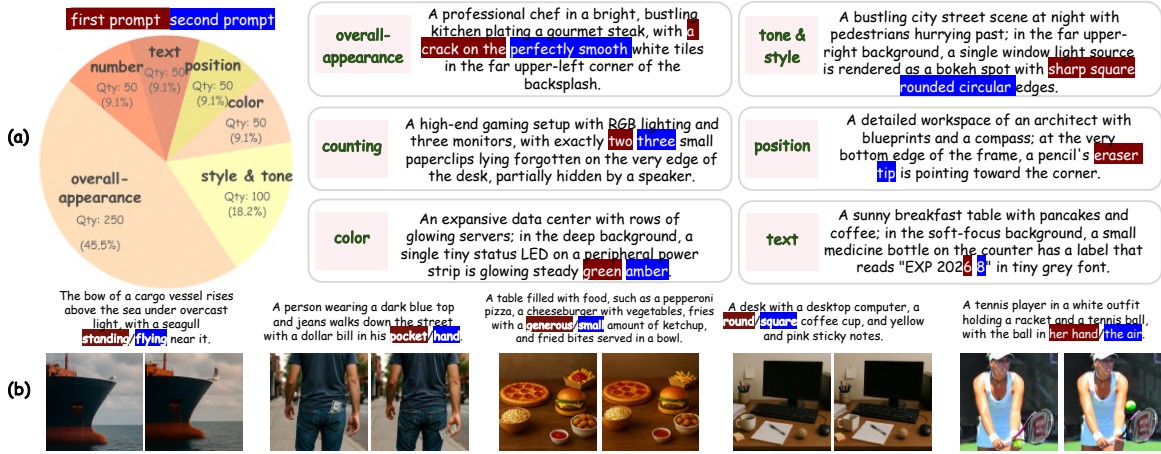

*Figure 2.* (a) Test case examples for each subtask in DeltaBench. (b) Examples of training data in `FocusDiff-Data`.

**Evaluation Protocols.** We employ Qwen3-VL-30B (Bai et al., 2025a) as the evaluation model to assess the fine-grained semantic alignment between generated images and prompts. Specifically, for each image-prompt pair, we query the VLM with binary (Yes/No) questions regarding both global alignment and minute-detail consistency. We record the probability of responding "yes" ("no") as $P_{yes}$ ($P_{no}$), with the semantic alignment score as $S(\mathcal{I}, \mathcal{T}) = P_{yes}/(P_{yes} + P_{no})$.

On this basis, given a subtask $\{(\mathcal{T}^{1,i}, \mathcal{T}^{2,i})\}$, for each prompt pair, we instruct a text-to-image model to generate corresponding images $\{\{(\mathcal{T}^{1,i}, \mathcal{I}_k^{1,i})\}_{k=1}^K, \{(\mathcal{T}^{2,i}, \mathcal{I}_k^{2,i})\}_{k=1}^K\}_{i=1}^N$, with each prompt generating $K = 2$ images. We define $s_k^{j,i} = S(\mathcal{I}_k^{j,i}, \mathcal{T}^{j,i})$, introduce two evaluation metrics: arithmetic mean $s_a = \frac{1}{4N} \sum_{i=1}^N \sum_{j=1}^2 \sum_{k=1}^2 s_k^{j,i}$, and geometric mean $s_g = \frac{1}{N} \sum_{i=1}^N \sqrt[4]{\prod_{j=1}^2 \prod_{k=1}^2 s_k^{j,i}}$. Here, $s_a$ measures the overall semantic alignment of the generated images with the prompts, while $s_g$ assesses the model's fine-grained precision and stability in generating images for similar prompts. $\mathcal{T}^{j,i}$ means the $j$-th prompt in $i$-th test case, while $\mathcal{I}_k^{j,i}$ means the $k$-th image generated for the $j$-th prompt in $i$-th test case.

## 3. Method: FocusDiff

In this section, we introduce FocusDiff, a novel text-to-image method that focuses on the differences between similar text-image pairs to enhance fine-grained text-image alignment. From the data perspective, we propose `FocusDiff-Data`, a large-scale dataset of paired samples to capture localized semantic shifts. From the training perspective, we further propose Pair-GRPO, an improved RL framework that extends GRPO to paired samples.

### 3.1. Data Perspective: `FocusDiff-Data`

While traditional datasets of text-image pairs effectively achieve global alignment, they often suffer from semantic ambiguity at the fine-grained level. The lack of granular annotations that explicitly map sentence constituents to specific image regions causes models to neglect subtle textual descriptions in favor of only dominant semantics. To bridge this gap without expensive manual annotation, we turn to differential learning.

We construct training instances as contrastive sets $\{(\mathcal{T}^1, \mathcal{I}^1), (\mathcal{T}^2, \mathcal{I}^2)\}$. While $\mathcal{T}^1$ and $\mathcal{T}^2$, as well as $\mathcal{I}^1$ and $\mathcal{I}^2$, are similar in overall expression and global semantics, they differ in fine-grained details that are often overlooked beyond the primary semantics. So $\mathcal{T}^1$ is semantically aligned with $\mathcal{I}^1$ but not with $\mathcal{I}^2$, and vice versa. In a AR context, this minimizes the distributional shift between samples, isolating the specific text tokens responsible for visual changes. It compels the model to learn a precise mapping: detecting how a modification in the textual condition $\Delta\mathcal{T}$ directly causes the variation in the visual output $\Delta\mathcal{I}$, thus achieving superior fine-grained control.

To obtain such paired data, especially pairs of similar images, we leverage a image-first pipeline and turn to image editing datasets (Wang et al., 2025c; Ye et al., 2025; Yu et al., 2024), which involve before-and-after-editing image pairs where only localized regions are modified. We collect image pairs covering diverse editing types to reflect differences in various attributes. Then we employ a powerful visual comprehension model to generate style-similar captions for each pair.

Specifically, we perform an initial screening of the collecting image pairs to assess three key aspects: (1) editing instructions following, (2) non-edited areas preserving, and (3) natural appearance. After eliminating substandard samples, we input the before-and-after image pair and the editing instruc-

tion into InternVL3.5-38B (Wang et al., 2025b), prompting it to generate captions with similar structure but different key words to highlight the subtle image differences.

After generating the captions $(\mathcal{T}^1, \mathcal{T}^2)$ for the images $(\mathcal{I}^1, \mathcal{I}^2)$, we then perform a post-verification to ensure three conditions: (1) $\mathcal{T}^1$ and $\mathcal{T}^2$ exhibit similar semantic structures; (2) each caption must align with its corresponding image; and (3) cross-alignment is prohibited. Any violations trigger regeneration and re-verification using InternVL3.5.

Ultimately, we retain about $500,000$ high-quality data pairs. Randomly selected examples from `FocusDiff-Data` are visualised in Figure 2(b), where the paired images or prompts exhibit only region-level or word-level differences. See more details in Appendix D.

### 3.2. Training Perspective: Pair-GRPO

With `FocusDiff-Data`, we first conduct a supervised text-to-image fine-tuning. Then we perform reinforcement learning based on an improved version of GRPO (Shao et al., 2024), as shown in Figure 3, realizing a better balance of exploration-exploitation trade-off.

**Vanilla GRPO for Image Generation.** We aim to adopt GRPO as the framework for reinforcement learning, which eliminates the value function and estimating the advantages in a group-relative manner. Specifically, for each prompt, we sample a group of $G$ outputs and compute the advantage $A_k$ by standardizing the individual reward $\mathtt{R}_{\mathcal{I}_k}$ against the group statistics, with a QA-based reward. The policy parameters are then updated via the following objective, incorporating a KL divergence penalty for stability:

$$
\mathcal{J}(\theta) = \mathbb{E}_{\substack{(\mathcal{T},a)\sim\mathcal{D} \\ \{y_i\}_{i=1}^{G} \sim \pi_{\theta_{\text{old}}}(\cdot|\mathcal{T})}} \left[ \frac{1}{\sum_{i=1}^{G}|y_i|} \sum_{i=1}^{G} \sum_{j=1}^{|y_i|} \left( \right. \right.
$$
$$
\left. \left. \min\left(\rho_{i,j}A_i, \text{clip}\left(\rho_{i,j}, 1-\varepsilon, 1+\varepsilon\right)A_i\right) - \beta D_{\text{KL}} \right) \right], \quad (1)
$$

where $\rho_{i,j}$ denotes the probability ratio between the new and old policies, and $D_{\text{KL}} = \frac{\pi_{ref}}{\pi} - \log\frac{\pi_{ref}}{\pi} - 1$ represents the KL divergence. See more detains in Appendix A.

**Pair-GRPO for Fine-Grained Semantic Focusing.** To enhance the model ability to capture subtle differences between two prompts, we extend the group concept in GRPO from images generated by a single prompt to those generated by a pair of similar prompts. This aligns with our core idea of comparing the outputs from similar prompts. Specifically, given a pair of input prompt $\{\mathcal{T}^1, \mathcal{T}^2\}$ with similar global expressions but fine-grained semantics differences, a group of $2G$ images $\{\mathcal{I}_k^1\}_{k=1}^{G}$ for $\mathcal{T}^1$ and $\{\mathcal{I}_k^2\}_{k=1}^{G}$ for $\mathcal{T}^2$ are sampled from the old policy. $\{\mathcal{I}_k^1\}_{i=1}^{G}$ and $\{\mathcal{I}_k^2\}_{k=1}^{G}$ are assigned to the same group $\mathcal{G}_0 = \{(\mathcal{T}^1, \mathcal{I}_k^1)\}_{k=1}^{G} \cup \{(\mathcal{T}^2, \mathcal{I}_k^2)\}_{k=1}^{G}$

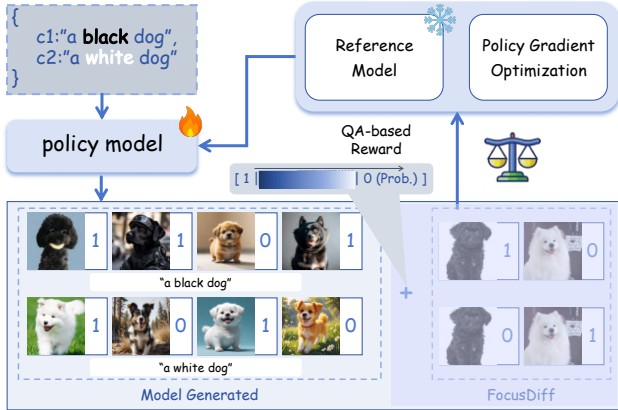

*Figure 3.* The framework of Pair-GRPO.

for advantage calculation.

Furthermore, from the `FocusDiff-Data` dataset, we could also obtain the ground-truth images $\hat{\mathcal{I}}^1$ and $\hat{\mathcal{I}}^2$ corresponding to $\mathcal{T}^1$ and $\mathcal{T}^2$. Despite the high similarity between $\hat{\mathcal{I}}^1$ and $\hat{\mathcal{I}}^2$, during construction we ensure that $\hat{\mathcal{I}}^1$ achieves a favorable reward score when conditioned on $\mathcal{T}^1$, but achieves an unfavorable score when conditioned on $\mathcal{T}^2$. Thus, if we further incorporate $\hat{\mathcal{I}}^1$ into the group, it can assume a dual role within the group: it serves as a positive guide in $\{(\mathcal{T}^1, \mathcal{I}_k^1)\}_{k=1}^{G}$ indicating the correct visual semantics, and as a cautionary counterexample in $\{(\mathcal{T}^2, \mathcal{I}_k^2)\}_{k=1}^{G}$, warning the model to avoid generating erroneous visual semantics that are commonly encountered. The same applies to $\hat{\mathcal{I}}^2$.

On this basis, we introduce a dynamic probability $p$ that starts at $1.0$ and gradually decreases to $0.0$ during RL training. At each training iteration, with probability $p$, we expand the group $\mathcal{G}$ to include the above additional pairs from `FocusDiff-Data`: $\mathcal{G} = \mathcal{G}_0 + \{(\mathcal{T}^1, \hat{\mathcal{I}}^1), (\mathcal{T}^1, \hat{\mathcal{I}}^2), (\mathcal{T}^2, \hat{\mathcal{I}}^1), (\mathcal{T}^2, \hat{\mathcal{I}}^2)\}$. Otherwise, the group remains as $\mathcal{G} = \mathcal{G}_0$. **This is a process of shifting focus from exploitation to exploration.** In the early stages of training, the labeled images from the dataset encourage more exploitation to the model, offering more appropriate guidance. As training progresses and the model's ability to grasp fine-grained differences strengthens, the probability of providing labeled images gradually decreases. And finally, we encourage model to develop advanced problem-solving strategies through fully autonomous exploration.

In each iteration, after defining the group concept, we employ the same way as vanilla GRPO to calculate the rewards, advantages and the objective function.

## 4. Experiments

We employ X-Omni (Geng et al., 2025) as the backbone, developing X-Omni-FocusDiff, excelling in text-to-image

*Table 1.* Results on DeltaBench with Qwen3-VL-30B as the evaluator. The best results within each category (diffusion or AR-based) are in **bold fonts** with the second best underlined.

| Method | #Params | Overall Appear. | | Color | | Counting | | Position | | Style&Tone | | Text | | **Average** | |
|---|---|---|---|---|---|---|---|---|---|---|---|---|---|---|---|
| | | $s_a \uparrow$ | $s_g \uparrow$ | $s_a \uparrow$ | $s_g \uparrow$ | $s_a \uparrow$ | $s_g \uparrow$ | $s_a \uparrow$ | $s_g \uparrow$ | $s_a \uparrow$ | $s_g \uparrow$ | $s_a \uparrow$ | $s_g \uparrow$ | $s_a \uparrow$ | $s_g \uparrow$ |
| *Diffusion Model* | | | | | | | | | | | | | | | |
| SD3 (Esser et al., 2024) | 8B | 68.4 | 60.3 | 89.8 | 87.5 | 48.2 | 39.4 | 45.1 | 34.2 | 82.3 | 75.7 | 83.4 | 77.9 | 69.5 | 62.5 |
| FLUX.1-dev (Labs, 2024) | 12B | 64.5 | 55.3 | 88.0 | 83.6 | 39.1 | 30.1 | 45.1 | 36.6 | 78.6 | 69.2 | 83.1 | 77.6 | 66.4 | 58.7 |
| Sana-1.5 (Xie et al., 2025a) | 4.8B | 68.2 | 60.2 | 90.8 | 88.6 | 40.4 | 33.2 | 43.0 | 32.1 | 87.5 | 83.4 | 61.2 | 54.0 | 65.2 | 58.6 |
| Flow-GRPO (Liu et al., 2025) | 2.5B | 68.6 | 60.1 | 90.0 | 87.8 | 41.8 | 29.2 | 44.5 | 33.2 | 83.8 | 78.7 | 83.8 | 78.4 | 68.8 | 61.2 |
| Lumina-Image 2.0 (Qin et al., 2025) | 2B | 62.2 | 51.9 | 83.3 | 78.3 | 35.1 | 20.4 | 44.1 | 32.7 | 84.9 | 79.8 | 42.1 | 28.7 | 58.6 | 48.6 |
| HiDream-I1-Full (Cai et al., 2025b) | 17B | 68.0 | 60.3 | 88.1 | 86.3 | 41.6 | 35.7 | 46.6 | 37.7 | 84.7 | 79.7 | 89.3 | 85.5 | 69.8 | 64.2 |
| HunyuanImage-2.1 (Team, 2025) | 17B | 67.6 | 60.3 | 80.1 | 76.3 | 60.4 | 54.7 | 59.1 | 49.1 | 81.2 | 78.2 | 41.4 | 32.8 | 65.4 | 58.6 |
| Z-Image (Cai et al., 2025a) | 6B | 66.5 | 61.2 | 86.5 | 86.8 | 46.5 | 39.8 | 46.0 | 36.4 | 88.5 | 82.2 | 94.5 | 92.8 | 71.8 | 66.5 |
| Qwen-Image (Wu et al., 2025b) | 7B+20B | 69.5 | 66.1 | 90.8 | 90.0 | 62.0 | 58.2 | 61.1 | 47.6 | 89.2 | 86.3 | 91.2 | 89.8 | 77.3 | 73.0 |
| FLUX.2-dev (Labs, 2025) | 24B+32B | 72.8 | 69.4 | 95.1 | 94.0 | 65.8 | 62.1 | 66.1 | 54.8 | 92.4 | 89.9 | 97.2 | 96.0 | 81.6 | 77.7 |
| *AR-based (AR+Diffusion or only AR)* | | | | | | | | | | | | | | | |
| SEED-X (Ge et al., 2024) | 17B | 47.1 | 32.9 | 71.3 | 62.4 | 34.4 | 24.8 | 21.4 | 10.3 | 79.0 | 71.2 | 22.0 | 13.5 | 45.9 | 36.1 |
| Show-o2 (Xie et al., 2025b) | 7B | 66.8 | 55.1 | 86.2 | 83.9 | 25.7 | 19.6 | 36.5 | 22.7 | 83.8 | 78.2 | 30.2 | 21.0 | 54.8 | 47.1 |
| Emu3 (Wang et al., 2024) | 8B | 58.7 | 46.2 | 75.2 | 64.9 | 36.1 | 26.2 | 34.4 | 26.2 | 80.1 | 71.7 | 28.0 | 17.6 | 52.1 | 42.2 |
| BLIP-3o (Chen et al., 2025a) | 8B | 66.3 | 57.7 | 86.5 | 82.4 | 36.8 | 31.0 | 46.0 | 34.0 | 86.8 | 82.8 | 44.1 | 31.0 | 61.2 | 53.2 |
| OmniGen2 (Wu et al., 2025c) | 7B | 68.1 | 58.6 | 86.9 | 81.9 | 36.1 | 28.6 | 47.1 | 40.1 | 83.8 | 77.7 | 75.3 | 68.0 | 66.0 | 58.7 |
| Bagel-Think (Deng et al., 2025) | 7B+7B | 65.9 | 55.9 | 84.9 | 81.3 | 40.7 | 31.9 | 43.4 | 35.8 | 80.8 | 71.6 | 39.5 | 28.9 | 59.5 | 50.9 |
| Janus-Pro (Chen et al., 2025b) | 7B | 65.4 | 56.6 | 86.8 | 84.5 | 25.5 | 17.0 | 44.2 | 25.9 | 85.3 | 81.0 | 49.1 | 37.9 | 59.4 | 50.6 |
| Lumina-mGPT 2.0 (Xin et al., 2025) | 7B | 59.0 | 48.7 | 78.5 | 70.2 | 28.5 | 21.7 | 32.1 | 24.1 | 78.1 | 68.8 | 30.4 | 19.3 | 51.1 | 42.1 |
| T2I-R1 (Jiang et al., 2025) | 7B | 68.5 | 61.6 | 90.4 | 87.6 | 37.1 | 28.6 | 46.3 | 35.4 | 87.0 | 82.7 | 66.8 | 59.2 | 65.8 | 58.9 |
| X-Omni (Geng et al., 2025) | 9.6B | 63.0 | 54.3 | 82.5 | 74.0 | 32.2 | 23.6 | 41.2 | 32.5 | 78.5 | 70.2 | 75.8 | 66.9 | 61.8 | 53.2 |
| GLM-Image (Team, 2026b) | 9B+7B | 67.2 | 60.8 | 89.2 | 86.2 | 48.2 | 40.2 | 47.1 | 37.2 | 89.8 | 82.8 | 95.8 | 92.9 | 72.9 | 66.7 |
| **X-Omni-FocusDiff (Ours)** | 7B+12B | 68.7 | 65.3 | 90.9 | 89.0 | 57.8 | 54.0 | 58.5 | 46.0 | 88.7 | 85.7 | 92.1 | 90.8 | 76.1 | 71.8 |

generation, with improved capabilities of vision-language alignment. More details are given in Appendix E and F.

### 4.1. Main Results on DeltaBench

We conduct zero-shot evaluations on DeltaBench for our model and recent advanced diffusion-based and AR-based text-to-image methods. We report the arithmetic mean scores $s_a$ and geometric mean scores $s_g$ in Table 1 with Qwen3-VL-30B (Bai et al., 2025a) as the evaluator. We have the following key findings of existing methods:

**(1) SOTA models in both categories exhibit strong alignment, driven by parameter scaling.** The top-performing models across both architectures, FLUX.2-dev and Qwen-Image (diffusion), and GLM-Image (AR-based) all achieve impressive arithmetic-mean scores. These results highlight a clear trend: **larger parameter counts generally correlate with superior fine-grained text-image alignment performance, regardless of the underlying architecture.**

**(2) Generation stability remains a major challenge for most models.** The gap between $s_g$ and $s_a$ reflects a model's generation stability. With the notable exceptions of FLUX.2-dev and Qwen-Image, most existing models exhibit significant instability with gaps exceeding 5.0 points, especially for AR-based models. For instance, the backbone X-Omni shows a 8.6-point gap. This suggests that precisely controlling fine-grained visual semantics remains difficult for models without specialized stability-enhancing training.

Compared to existing methods and the backbone, X-Omni-

FocusDiff achieves the following advantages: **(1) Significantly improved text-image alignment.** After training, X-Omni-FocusDiff achieves superior vision-language alignment, reaching an average $s_a$ of 76.1. Compared to the backbone X-Omni, our model achieves a massive improvement of 14.3 points in $s_a$ and 18.6 points in $s_g$. Remarkably, our model outperforms GLM-Image and comes close to the performance of the 27B Qwen-Image, demonstrating that our FocusDiff method allows a mid-sized model to reach the alignment capabilities typically reserved for much larger SOTA models.

**(2) Enhanced Generation Stability matching top-tier SOTA.** Our method effectively bridges the stability gap inherent in AR-based generation. X-Omni-FocusDiff reduces the $s_a - s_g$ gap to only 4.3 points, making it one of the few models, alongside FLUX.2-dev and Qwen-Image, to achieve a stability gap of less than 5 points. This is a significant enhancement compared to the backbone's 8.6-point gap, which allows the model to reliably focus on subtle semantic differences for stable, high-quality image generation.

### 4.2. Main Results on Existing Benchmarks

We further conduct zero-shot evaluation on 3 text-to-image benchmarks: GenEval (Ghosh et al., 2023), LongText-Bench (Geng et al., 2025), and DPG-Bench (Hu et al., 2024). The comparison against diffusion and AR-based models is shown in Table 2. We have the following observations: **(1)** In most settings, our model achieves superior performance than various leading models. Notably, on GenEval, X-Omni-

*Table 2.* Comparison with leading models on GenEval, LongText-Bench and DPG-Bench on zero-shot text-to-image generation. The best results within each category (diffusion or AR-based) are in **bold fonts** with the second best underlined.

| Method | #Params | Overall↑ | Single Obj.↑ | Two Obj.↑ | Counting↑ | Color↑ | Position ↑ | Color Attr. ↑ | English↑ | Average↑ |
|---|---|---|---|---|---|---|---|---|---|---|
| | | | | | **GenEval** | | | | **LongText-Bench** | **DPG-Bench** |
| *Diffusion-Only* | | | | | | | | | | |
| SD3 (Esser et al., 2024) | 8B | 0.74 | 0.99 | 0.94 | 0.72 | 0.89 | 0.33 | 0.60 | - | 84.08 |
| FLUX.1-dev (Labs, 2024) | 12B | 0.66 | 0.98 | 0.79 | 0.73 | 0.77 | 0.22 | 0.45 | 0.607 | 83.79 |
| Sana-1.5 (Xie et al., 2025a) | 4.8B | 0.81 | 0.99 | 0.93 | 0.86 | 0.84 | 0.59 | 0.65 | - | 84.70 |
| Lumina-Image 2.0 (Qin et al., 2025) | 2B | 0.78 | 0.99 | 0.87 | 0.67 | 0.86 | 0.70 | 0.62 | - | 82.32 |
| HunyuanImage-2.1 (Team, 2025) | 17B | 0.79 | 0.98 | 0.92 | 0.71 | 0.86 | 0.66 | 0.61 | - | 85.15 |
| Z-Image (Cai et al., 2025a) | 6B | 0.84 | **1.00** | 0.94 | 0.78 | **0.93** | 0.62 | 0.77 | 0.935 | 88.14 |
| Qwen-Image (Wu et al., 2025b) | 7B+20B | **0.91** | **1.00** | **0.95** | 0.93 | 0.92 | **0.87** | **0.83** | **0.943** | **88.32** |
| *AR-based (AR+Diffusion or only AR)* | | | | | | | | | | |
| SEED-X (Ge et al., 2024) | 17B | 0.49 | 0.96 | 0.57 | 0.29 | 0.82 | 0.14 | 0.15 | - | - |
| Show-o2 (Xie et al., 2025b) | 7B | 0.76 | **1.00** | 0.87 | 0.58 | **0.92** | 0.52 | 0.62 | 0.006 | 86.14 |
| Emu3 (Wang et al., 2024) | 8B | 0.54 | 0.98 | 0.71 | 0.34 | 0.81 | 0.17 | 0.21 | - | 80.60 |
| BLIP-3o (Chen et al., 2025a) | 8B | 0.84 | - | - | - | - | - | - | 0.021 | 81.60 |
| OmniGen2 (Wu et al., 2025c) | 7B | 0.80 | **1.00** | 0.95 | 0.64 | 0.88 | 0.55 | 0.76 | 0.561 | 83.57 |
| Bagel-Think (Deng et al., 2025) | 14B | 0.82 | 0.99 | 0.94 | 0.81 | 0.88 | 0.64 | 0.63 | 0.373 | 85.07 |
| Janus-Pro (Chen et al., 2025b) | 7B | 0.80 | 0.99 | 0.89 | 0.59 | 0.90 | 0.79 | 0.66 | 0.019 | 84.17 |
| Lumina-mGPT 2.0 (Xin et al., 2025) | 7B | 0.80 | **1.00** | 0.92 | 0.57 | 0.88 | 0.70 | 0.72 | - | 79.11 |
| T2I-R1 (Jiang et al., 2025) | 7B | 0.79 | 0.99 | 0.91 | 0.53 | 0.91 | 0.76 | 0.65 | - | 84.42 |
| X-Omni (Jiang et al., 2025) | 7B+12B | 0.83 | 0.98 | 0.95 | 0.75 | 0.91 | 0.71 | 0.68 | 0.900 | 87.65 |
| GLM-Image (Team, 2026b) | 9B+7B | 0.89 | 0.99 | 0.94 | 0.89 | 0.89 | 0.82 | **0.81** | 0.952 | 84.78 |
| GPT-4o (OpenAI, 2024) | - | 0.85 | 0.99 | 0.92 | 0.85 | 0.91 | 0.75 | 0.66 | **0.956** | 86.23 |
| **X-Omni-FocusDiff (Ours)** | 7B+12B | **0.90** | **1.00** | **0.96** | **0.91** | 0.91 | **0.83** | **0.81** | 0.935 | **88.35** |

FocusDiff achieves higher overall score than large-scale parameter models like Qwen-Image and proprietary models like GPT-4o. This underscores that we endow the MLLM with enhanced capability of vision-language alignment. **(2)** Compared to the backbone model X-Omni, our method achieves consistent improvements across all benchmarks, significantly enhancing the capabilities of base model with strong effectiveness.

### 4.3. Qualitative Examples

**Image Generation with Similar Prompts.** Figure 4 present a direct qualitative comparison among X-Omni, Nano Banana Pro (Google, 2025b) and our model on semantically similar prompts (two samples per prompt). We can see that X-Omni struggles to precisely control the fine-grained requirements of similar prompts. While even the leading Nano Banana Pro occasionally yields results inconsistent with the prompt. In contrast, X-Omni-FocusDiff accurately discerns subtle semantic distinctions, consistently generating high-quality images aligning with the prompts. More Qualitative Examples of the generated images are given in **Figure 9** in Appendix G.

**Image Generation with Counterfactual Prompts.** Endowing our model with fine-grained control over visual details enables it to generate images that accurately match counterfactual prompts that are rarely found in real-world, as shown in Figure 6. For instance, given "A giant hamburger aims to eat a small person on a plate", though X-Omni captures the entity scales, it fails to distinguish the subject-object relationship. In contrast, our model accurately resolves this semantic ambiguity, correctly depicting the hamburger as the active agent that aims to eat the person.

**Qualitative Examples Generated on Similar Prompts within DeltaBench.** Detailed qualitative examples on DeltaBench for our model are presented in **Appendix Figure 10** due to page limitations. These examples highlight a key strength of our model: beyond aligning with the main subject semantics, our model demonstrates precise control over intricate, off-center details. Whether it is the specific geometry of background bokeh, the text on a corner bottle's label, or the color of tiny distant lights, our model handles these fine-grained constraints with remarkable accuracy.

### 4.4. In-depth Analysis

**Effect of Individual Components.** To investigate the effectiveness of each component, we trained the following ablation models: **(1) w/o Group Expanding:** The group concept is restricted to images generated from a single prompt. **(2) w/o GT Image:** We set $p = 0.0$ and do not provide ground-truth images during RL. **(3) Vanilla GRPO:** We fully degrade Pair-GRPO to the vanilla GRPO. **(4) w/o `FocusDiff-Data`:** We select a set of commonly-used prompts to replace `FocusDiff-Data` for Vanilla GRPO training. As shown **Rows3-5** in Table 3, Pair-GRPO consistently outperforms other ablated algorithms on DeltaBench. Moreover, as shown in **Rows5-6**, the performance obtained from training with `FocusDiff-Data` outperforms that with commonly-used prompts. This indicates that both Pair-GRPO and `FocusDiff-Data` enable the model to effectively focus on the fine-grained prompt requirements, thus achieving better text-image alignment on DeltaBench.

**Stricter Evaluation on GenEval with FPR.** We introduce a stricter evaluation metric on GenEval termed Full-Pass Rate (FPR). FPR is defined as the success rate of prompts, where a prompt is deemed successful only if every

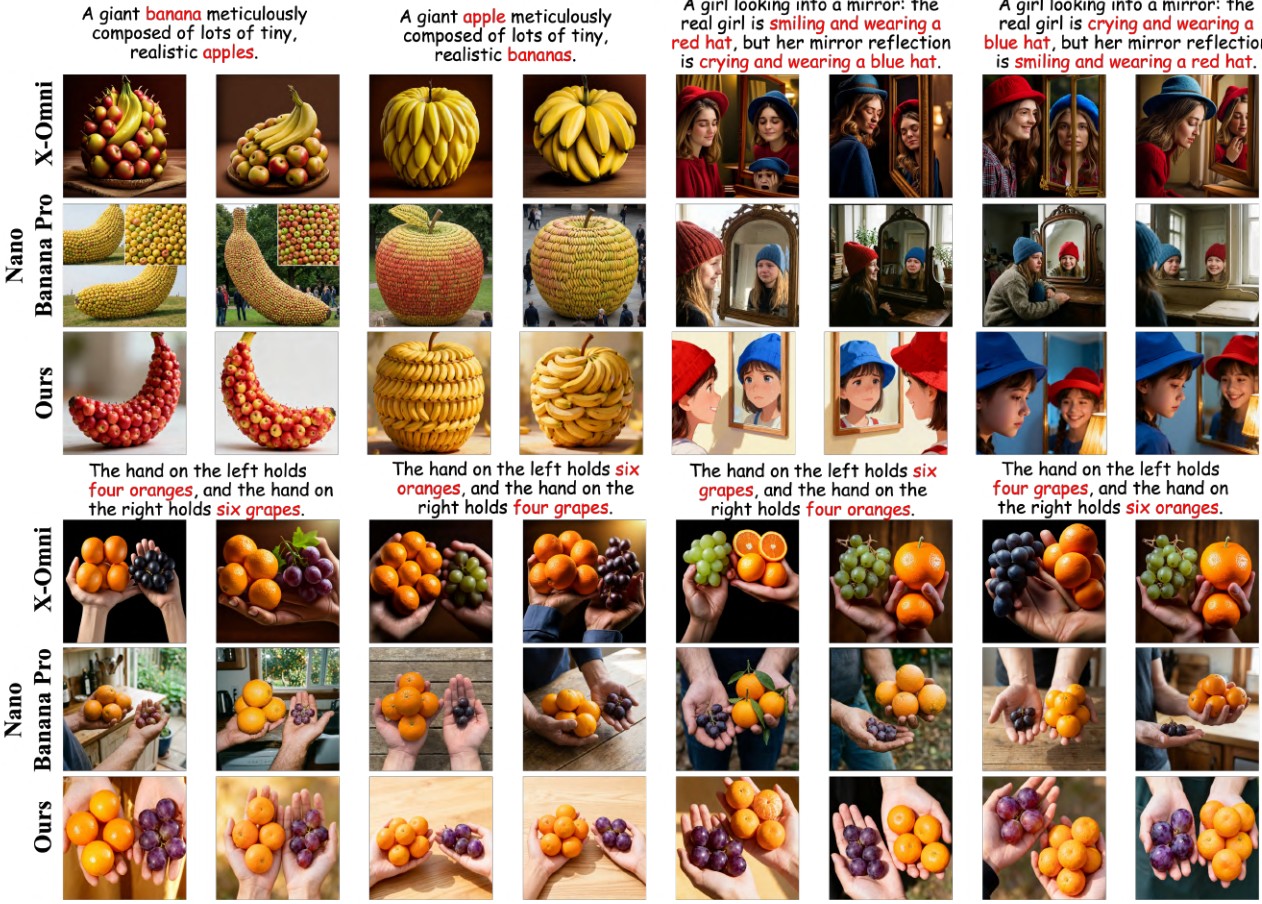

*Figure 4.* Qualitative Comparisons of X-Omni, Nano Banana Pro and our model on similar prompts (generating two images each prompt).

image generated for that prompt is correct. As shown in Figure 5(a), X-Omni and GLM-Image suffer significant performance drops of 9 and 15 points, respectively, in FPR. In contrast, our model achieves a substantially higher FPR with a marginal decline of only 4 points, demonstrating a higher success rate and superior stability in image generation.

**Stricter Evaluation on DeltaBench with Extended Sampling.** We also conduct stricter evaluation on DeltaBench by gradually extending the number of images sampled per prompt from 2 to 8, with the resulting performance trends illustrated in Figure 5(b). For X-Omni and GLM-Image, increasing the number of sampled images leads to a significant decline in the geometric mean, indicating poor generation stability. In contrast, for our model, the geometric mean decreases only very slightly (from 71.8 to 69.8) as the number of generated samples increases. These confirm that our model consistently generates high-quality images with enhanced stability.

**Stability of Pair-GRPO Training** As acknowledged in (Yu et al., 2025; Xiong et al., 2025), GRPO is notoriously difficult to train due to its inherent instability. However,

we find that Pair-GRPO not only surpasses vanilla GRPO in performance, but also enhances the training stability. In Figure 5(c), we present the reward trends at different training steps for both vanilla GRPO and Pair-GRPO. Compared with the fluctuating rewards of vanilla GRPO, Pair-GRPO demonstrates a steady reward improvement, which suggests that it effectively optimizes the training instability.

**Choice of Evaluation Models.** To validate the reliability of VLM-based evaluation, we selected 70 human-annotated images

*Table 4.* Pearson-r with human eval.

| Model | Pearson-r↑ |
|---|---|
| Qwen3-VL-30B (Bai et al., 2025a) | **0.77** |
| Qwen2.5-VL-72B (Bai et al., 2025b) | 0.71 |
| InternVL3.5-38B (Wang et al., 2025b) | 0.76 |
| GLM4.6v-flash-9B (Team, 2026a) | 0.69 |
| Gemini3-Pro (Google, 2025a) | 0.83 |

and employed several models—including Qwen2.5-VL, Qwen3-VL, InternVL3.5, GLM4.6v-flash, Gemini3-Pro for assessment. We then calculated the Pearson correlation coefficients to quantify the alignment between each VLM's output and human judgment. As illustrated in Table 4, Qwen3-VL-30B and InternVL3.5-38B achieved the highest correlations among the open-source models. Consequently, we selected InternVL3.5-38B as the reward model for RL

*Table 3.* Ablation Study on DeltaBench with Qwen3-VL-30B as the evaluator.

| Method | Overall Appear. | | Color | | Counting | | Position | | Style&Tone | | Text | | **Average** | |
|---|---|---|---|---|---|---|---|---|---|---|---|---|---|---|
| | $s_a \uparrow$ | $s_g \uparrow$ | $s_a \uparrow$ | $s_g \uparrow$ | $s_a \uparrow$ | $s_g \uparrow$ | $s_a \uparrow$ | $s_g \uparrow$ | $s_a \uparrow$ | $s_g \uparrow$ | $s_a \uparrow$ | $s_g \uparrow$ | $s_a \uparrow$ | $s_g \uparrow$ |
| (1) **X-Omni** | 63.0 | 55.4 | 82.5 | 75.1 | 32.2 | 24.7 | 41.2 | 33.6 | 78.5 | 71.3 | 75.8 | 68.0 | 61.8 | 54.3 |
| (2) **X-Omni-FocusDiff** | **68.7** | **65.3** | **90.9** | **89.0** | **57.8** | **54.0** | **58.5** | **46.0** | **88.7** | **85.7** | **92.1** | **90.8** | **76.1** | **71.8** |
| (3) w/o Group Expanding | 66.5 | 62.2 | 89.2 | 86.8 | 55.4 | 49.1 | 54.2 | 40.5 | 86.3 | 81.5 | 89.4 | 87.3 | 73.5 | 67.9 |
| (4) w/o GT Image | 67.4 | 63.9 | 89.8 | 87.9 | 57.7 | 52.2 | 57.1 | 43.2 | 87.6 | 83.7 | 91.0 | 89.1 | 75.1 | 70.0 |
| (5) Vanilla GRPO | 66.2 | 61.5 | 88.9 | 85.4 | 52.1 | 45.8 | 51.4 | 38.6 | 85.4 | 80.1 | 87.5 | 84.6 | 71.9 | 66.0 |
| (6) w/o `FocusDiff-Data` | 63.8 | 56.2 | 83.1 | 75.8 | 33.5 | 25.2 | 42.1 | 34.0 | 79.2 | 72.1 | 76.5 | 69.2 | 63.0 | 55.4 |

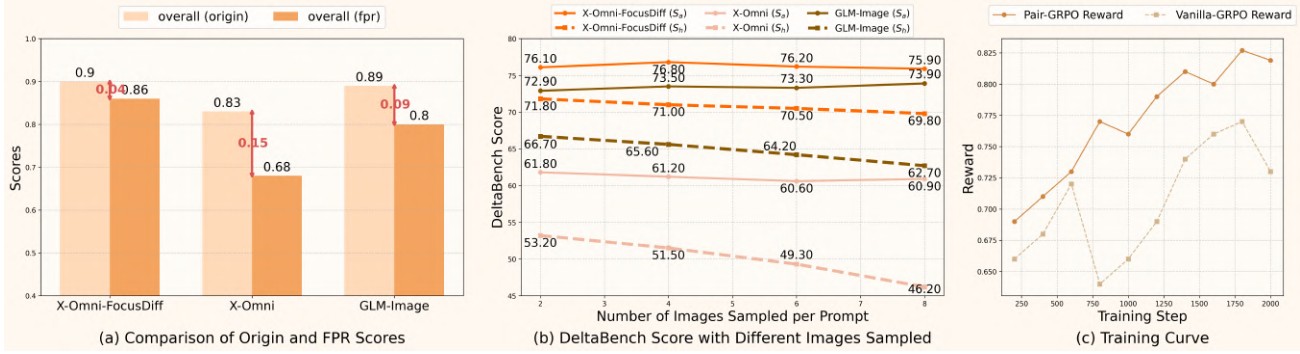

*Figure 5.* (a) Evaluation on GenEval with FPR overall score and original overall score. (b) Evaluation on DeltaBench when extending the number of images sampled per prompt from 2 to 8. (c) Reward trend for vanilla GRPO training and Pair-GRPO training.

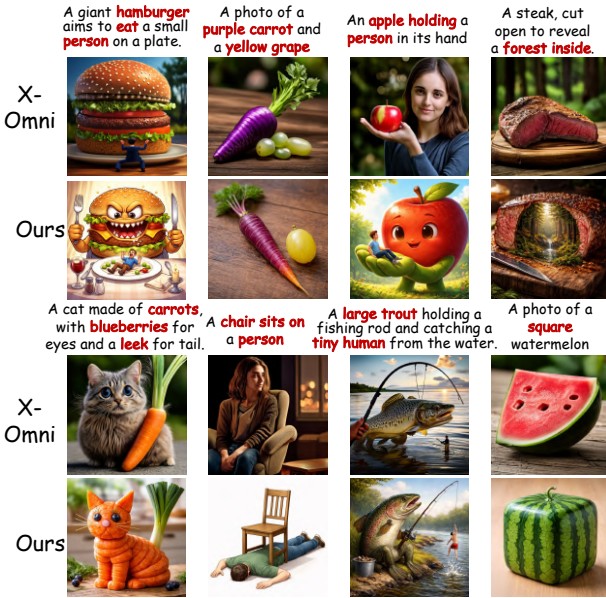

*Figure 6.* Image Generation with Counterfactual Prompts.

training, while reserving Qwen3-VL-30B for benchmark evaluation to mitigate the risk of reward hacking.

## 5. Related Work

While diffusion models (Labs, 2024; Esser et al., 2024) have long dominated visual generation, the autoregressive (AR) paradigm has recently surged in prominence, extending the scalability of Large Language Models to the vi-

sual domain (Google, 2025b; OpenAI, 2024). These approaches (Geng et al., 2025; Team, 2026b; Pan et al., 2024; 2025b; 2026b; Team, 2026b; Chen et al., 2025b) typically encode images into discrete tokens via a visual encoder such as SigLIP (Tschannen et al., 2025), formulating generation as a next-token prediction task. And the predicted tokens are subsequently decoded back into images via a VQ-VAE (Esser et al., 2021) generator or a diffusion decoder. Notably, AR is inherently amenable to reinforcement learning (Wang et al., 2025a), satisfying optimality conditions for policy improvement akin to LLM practices (Guo et al., 2025; Jiang et al., 2025). Whereas prior works primarily prioritize global semantic coherence, FocusDiff leverages this RL potential to address the critical gap in fine-grained text–to-image alignment, enabling precise control over visual tokens.

Regarding datasets, some recent efforts also introduce paired-prompt datasets to assess compositional generation. However, predecessors like Winoground-T2I (Zhu et al., 2023) and EvoGen (Han et al., 2024) focus on major semantic shifts and suffer from poor image consistency due to prompt-first construction. In contrast, we propose DeltaBench and `FocusDiff-Data` to target fine-grained, localized semantics. Crucially, `FocusDiff-Data` leverages an image-first pipeline, creating strictly aligned image pairs that enable models to learn precise control over minute visual tokens. See more details in Appendix B.

# 6. Conclusion

In this paper, we propose **DeltaBench**, a benchmark revealing that existing models struggle with fine-grained text-image alignment in text-to-image generation. To bridge this gap, we introduce **FocusDiff**, a training paradigm with a novel dataset and an improved RL algorithm, enhancing fine-grained text-image semantic alignment by focusing on subtle differences between similar text-image pairs. On this basis, we develop X-Omni-FocusDiff, significantly outperforms most prior prominent methods on DeltaBench and existing benchmarks.

# Acknowledgements

This work was supported by the National Key R&D Program of China (2025ZD0123100), the NSFC (62436007), the Key R&D Program in Zhejiang Province (No. 2025C01030), the Zhejiang NSF (LRG25F020001), the Ministry of Culture and Tourism (No.2023DMKLB002), and Ant Group.

# Impact Statement

**Ethical Impacts**. This study does not raise any ethical concerns. The research does not involve subjective assessments or the use of private data. Only publicly available datasets are utilized for experimentation.

**Expected Societal Implications**. The ability to generate hyper-realistic imagery raises significant concerns regarding the spread of misinformation and public deception. To mitigate these risks, it is imperative to establish stricter regulatory frameworks. Furthermore, implementing robust detection mechanisms and digital watermarking is essential to verify content authenticity and prevent malicious misuse. And the issue highlighted is not unique to our method but is prevalent across different techniques for text-to-image generation.

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

# Appendix Overview

In this supplementary material, we present:

- Vanilla GRPO for Autoregressive Image Generation in Section A.

- Compared with Prior Compositional Text-to-Image Benchmarks and Datasets in Section B

- More Details on DeltaBench in Section C.

- More Details on `FocusDiff-Data` in Section D.

- Implementation Details in Section E.

- Evaluation Details in Section F.

- More Experimental Results in Section G.

# A. Vanilla GRPO for Autoregressive Image Generation.

We adopt Group Relative Policy Optimization (GRPO) as the framework for reinforcement learning, GRPO enhances PPO by eliminating the value function and estimating the advantages in a group-relative manner. Specifically, given the input prompt $\mathcal{P}$, the old policy $\pi_{\theta_{old}}$ first samples a group of $G$ individual images as the response group $\mathcal{G} = \{\mathcal{I}_i^1\}_{i=1}^G$. We input each response with the group into the reward function to obtain the individual reward $\mathtt{R}_{\mathcal{I}_i}$. We then calculate the advantages $\{A_i\}_{i=1}^G$, where each $A_i$ measure the relative quality of output compared to the average reward:

$$A_i = \frac{\mathtt{R}_{\mathcal{I}_i} - \mathrm{mean}\big(\{\mathtt{R}_{\mathcal{I}_i}\}_{i=1}^G\big)}{\mathrm{std}\big(\{\mathtt{R}_{\mathcal{I}_i}\}_{i=1}^G\big)} \quad (2)$$

Then, we update the policy network parameters by the following training loss:

$$\mathcal{J}(\theta) = \mathbb{E}_{\substack{(\mathcal{P},a)\sim\mathcal{D} \\ \{y_i\}_{i=1}^G \sim \pi_{\theta_{old}}(\cdot|\mathcal{P})}} \left[ \frac{1}{\sum_{i=1}^G |y_i|} \sum_{i=1}^G \sum_{j=1}^{|y_i|} \Big( \right.$$
$$\left. \min\Big(\rho_{i,j}A_i, \mathrm{clip}\big(\rho_{i,j}, 1-\varepsilon, 1+\varepsilon\big)A_i\Big) - \beta D_{\mathrm{KL}}\Big)\right], \quad (3)$$

where $D_{\mathrm{KL}} = \frac{\pi_{ref}}{\pi} - \log\frac{\pi_{ref}}{\pi} - 1$ is the the KL divergence to maintain training stability. And $\rho_{i,j} = \frac{\pi_\theta(y_{i,j}|\mathcal{P},y_{i,<j})}{\pi_{\theta_{old}}(y_{i,j}|\mathcal{P},y_{i,<j})}$ is the ratio between probabilities of $\pi_\theta$ and $\pi_{\theta_{old}}$ for outputting current token. As for reward calculation, the overall design philosophy of our reward model is to leverage QA-based visual comprehension models (e.g., InternVL3.5 Wang et al., 2025b), which will return a consistency score from 0 to 1 for each text-image pair, to assess the semantic accuracy of the generated image.

# B. Compared with Prior Compositional Text-to-Image Benchmarks and Datasets

Some recent efforts have also introduce paired-prompt datasets to assess compositional generation. For example, Winoground-T2I (Zhu et al., 2023) is a compositional evaluation benchmark where each case has paired prompts, while EvoGen (Han et al., 2024) is dedicated to constructing contrastive pairs for training. While these approaches share some similarities with ours, our method differs in several fundamental aspects. To elucidate the essential distinctions between our work and existing paired-prompt datasets, we select Winoground-T2I and EvoGen as representative baselines. Specifically, we compare DeltaBench against Winoground-T2I, and `FocusDiff-Data` against EvoGen.

## B.1. Compare DeltaBench and Winoground-T2I

Though both benchmarks feature test cases of paired prompts that appear similar, they differ in design motivation.

The contrastive pairs in **Winoground-T2I** are specifically designed for compositional evaluation. While there may only be a difference of one or two words between the two prompts, the main semantic meaning of the prompts is different, leading to the target images with a drastic variation. Here are two typical examples:

- *"Real police officer with real police car" vs. "Toy police officer with toy police car"*.

- *"A boy jumping towards the fence and the river" vs. "A boy jumping away from the fence and the river"*.

As shown, the distinguishing words between the prompts are modifiers of the core subject's semantics. The difference between "toy" and "real" determines overall style of the main scene, and the difference between "towards" and "away" dictates the subject's positional orientation. Consequently, the resulting target images are vastly different, which allows Winoground-T2I to evaluate whether the model has learned the semantics of these core modifying words.

**In contrast, DeltaBench** not only builds upon the compositional evaluation, but also assesses whether the images generated by the model attend to fine-grained semantic details that are often overlooked beyond the primary semantics of a prompt. For instance, consider the following two prompts:

- *"A man in a blue jacket with buttons holds a coffee cup in a park." vs. "A man in a blue jacket with a zipper holds a coffee cup in a park."*

- *"A sunny breakfast table with pancakes and coffee; in the soft-focus background, a small medicine bottle on*

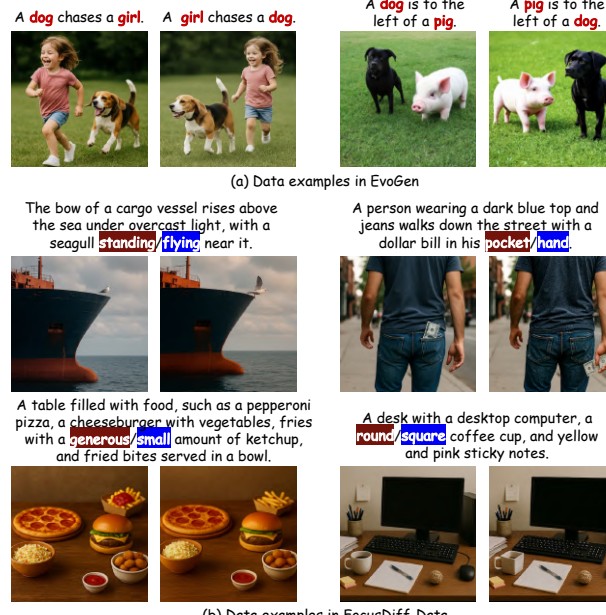

(a) Data examples in EvoGen

(b) Data examples in FocusDiff-Data

*Figure 7.* We present some typical data examples in EvoGen and `FocusDiff-Data`. Compared to EvoGen, `FocusDiff-Data` not only builds upon compositional learning of relational terms, but also emphasizes fine-grained semantic details that are often overlooked beyond the primary semantics.

> *the counter has a label that reads "EXP 2026" in tiny grey font." vs. "A sunny breakfast table with pancakes and coffee; in the soft-focus background, a small medicine bottle on the counter has a label that reads "EXP 2028" in tiny grey font."*

The core semantic content of both prompts is largely identical, depicting a man in a blue jacket holding a coffee cup in a park. However, a subtle, fine-grained difference exists concerning whether the jacket is fastened with buttons or a zipper. Thus, DeltaBench requires models to generate the correct subject semantics while also attending to these subtle details and correctly generating these easily overlooked aspects.

## B.2. Compare `FocusDiff-Data` and EvoGen

Although both `FocusDiff-Data` and EvoGen appear dedicated to constructing contrastive pairs, they differ in the following fundamental aspects, with the qualitative comparisons shown in Figure 7.

**Motivations and Design Purpose.** The contrastive pairs utilized in **EvoGen** are primarily designed to permute the core attributes or relational states of entities to facilitate compositional learning. Typical instances involve swapping the subject and object while preserving the predicate, as demonstrated by examples such as:

- *"A girl chases a dog" vs. "A dog chases a girl."*

- *"A dog is to the left of a pig" vs. "A pig is to the left of a dog."*

These pairs effectively compel a model to acquire the semantics of relational terms, such as "chase" and "left", which are pivotal to the overall meaning of a text prompt. Although prompts like "A girl chases a dog" and "A dog chases a girl" only differ in word order, their core semantic roles are reversed, resulting in corresponding ground-truth images that depict clearly distinct visual narratives.

**In contrast, `FocusDiff-Data`** not only builds upon the compositional learning of relational terms, but also emphasizes fine-grained semantic details that are often overlooked beyond the primary semantics of a prompt. For instance, consider the following two prompts:

- *"A tennis player in a white outfit holding a racket and a tennis ball, with the ball in her hand."*

- *"A tennis player in a white outfit holding a racket and a tennis ball, with the ball in the air."*

The core semantic content of both prompts is largely identical, depicting a tennis player engaged in playing tennis. However, a subtle, fine-grained difference exists concerning whether the ball is still in the hand or has been tossed into the air. As the capabilities of generative models continue to advance, it becomes increasingly crucial to capture and control subtle details within the prompts for generating images, especially the minor details beyond the core semantics, which is a key motivation for the design of our `FocusDiff-Data`.

**In summary**, EvoGen aims for the model to learn the compositional semantics, such as relational terms, which constitute the core semantics of the prompt. While FocusDiff builds upon this by also encouraging the model to attend to minor, easily overlooked details beyond the core semantics, thus ensuring the generated image perfectly aligns with the prompt's requirements.

**Construction Pipeline and Data Characteristics.** Driven by divergent goals, EvoGen and `FocusDiff-Data` adopt fundamentally different data-construction pipelines, which in turn shape the distinct characteristics of their respective contrastive pair data.

As **EvoGen** is designed for the compositional learning of core semantics—such as entity attributes or relational states—local consistency (e.g., subject identity, background details) is therefore irrelevant for its training objective. The sole requirement is that the two generated images faithfully

render the core semantic contrast encoded in the paired prompts.

Consequently, EvoGen follows a prompt-first paradigm. The process begins by engineering semantically contrasting prompt pairs and then synthesizing the corresponding images using generative models (e.g., SD3). However, it often results in poor consistency between the two images in a pair, such as the background and subject IP attributes.

**In contrast, FocusDiff** targets the control of fine-grained semantics, thus necessitating the model to infer how subtle changes in textual tokens lead to specific, minute changes in the visual images. This demands near-identical images whose only deviations are localized and subtle.

We therefore implement an image-first pipeline. We collect numerous pairs from existing image editing datasets featuring before-and-after-editing images that only exhibit localized, subtle differences. We then generate style-consistent captions for these pairs with differing words to highlight the subtle image differences.

**In summary**, EvoGen employs a prompt-first pipeline, resulting in significant inconsistencies between the two images within a pair. In contrast, `FocusDiff-Data` utilizes an image-first pipeline, where the two images within a pair only exhibit localized, subtle differences with strong consistency.

## C. More Details on DeltaBench

Each test case in DeltaBench contains two similar prompts with subtle differences. The two prompts exhibit word-level differences that lead to distinctions in six types of fine-grained semantic aspects: (1) Overall appearance difference; (2) Color difference; (3) Counting difference; (4) Position difference; (5) Style & Tone difference; (6) Text difference. Next, we will provide a detailed explanation of these types.

- **Color:** The prompt pair shares the exact same subject and background description, differing only in the color adjective assigned to a specific object. For example, a status LED is described as "green" in the first prompt versus "amber" in the second, while the rest of the server room context remains frozen.

- **Position:** The two prompts maintain the same set of objects and scene layout. The variation is strictly limited to the preposition or directional phrase describing the relative orientation of a specific item. For instance, a pencil eraser is pointing "toward" the corner versus "away from" the corner, without altering the position of the desk or blueprints.

- **Text:** The visual content and composition of the scene are anchored. The only difference lies in the specific quote or numeric sequence displayed on a text-bearing

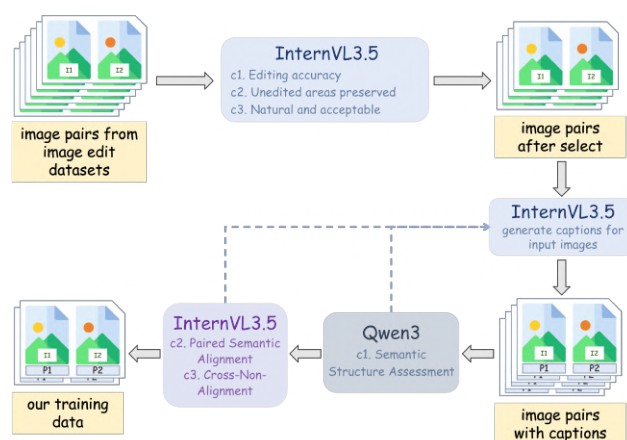

*Figure 8.* The pipeline for constructing `FocusDiff-Data`.

surface. For example, a medicine label in corner reads "EXP 2026" in one version and "EXP 2028" in the other, preserving the label's font, size, and placement.

- **Style & Tone:** Instead of altering the global semantic content, this category focuses on swapping specific adjectives that control local artistic effects or atmosphere. For example, background bokeh is described with "hexagonal" edges versus "circular" edges, while the foreground subject and action remain strictly invariant.

- **Counting:** The prompts are duplicates except for a single numerical adjective modifying a specific object class. This tests the model's ability to adjust the quantity (e.g., "two" paperclips vs. "three" paperclips) locally, without hallucinating changes to the surrounding clutter or the object's type.

- **Overall-appearance:** The subject's identity and category remain unchanged. The difference is confined to an adjective or phrase describing a fine-grained physical detail or texture. For instance, a tile surface is described as having a "tiny hairline crack" versus being "perfectly smooth," ensuring the kitchen context itself is not re-imagined.

## D. More Details on `FocusDiff-Data`

In this section, we give more details on how to construct `FocusDiff-Data` from the image editing dataset (Wang et al., 2025c; Ye et al., 2025), with the pipeline shown in Figure 8. In the first step, we conduct data cleaning on the raw data to retain only high-quality samples. Using the InternVL3.5-38B model (Wang et al., 2025b), providing it with the before-after-editing images and the editing instruction, we evaluate three key aspects with the following prompts: *(1) whether the edited image follows the editing instructions*; *(2) whether the non-edited areas of the edited*

*image remain consistent with the original image*; and *(3) whether the overall quality and natural appearance of the edited image are acceptable*. We filter out any pairs that fail to meet these criteria.

Subsequently, we input the pair of before-and-after images along with the editing instructions into InternVL3.5-38B (Wang et al., 2025b). We prompt it to generate a pair of captions for the images that share a similar stylistic structure but differ only in individual words, thereby highlighting the differences between the images.

After generating the captions $(\mathcal{P}_1, \mathcal{P}_2)$ for the images $(\mathcal{I}_1, \mathcal{I}_2)$, we conduct a post-verification operation with three conditions: **(1)** Using the Qwen3 model (Yang et al., 2025), we assess whether $\mathcal{P}_1$ and $\mathcal{P}_2$ exhibit similar semantic structures; **(2)** Using the InternVL3.5-38B model (Wang et al., 2025b), we verify whether $\mathcal{P}_1$ and $\mathcal{I}_1$, as well as $\mathcal{P}_2$ and $\mathcal{I}_2$, were semantically aligned. **(3)** We further leverage InternVL3.5-38B to ensure that $\mathcal{P}_1$ and $\mathcal{I}_2$, as well as $\mathcal{P}_2$ and $\mathcal{I}_1$, are not semantically aligned. If all of three conditions are satisfied, the sample is deemed valid and included in our training dataset. Otherwise, we request the InternVL3.5-38B to regenerate captions for the two images and conduct the post-verification again. If the post-verification still fails, the image pair is then discarded. Finally, we retained approximately $500,000$ high-quality data pairs.

## E. Implementation Details

**Supervised Fine-Tuning.** We first leverage `FocusDiff-Data` to conduct text-to-image supervised fine-tuning on X-Omni (Geng et al., 2025). The objective function is $p(y) = \frac{1}{S} \sum_{i=1}^{S} \log P_\theta(y_i | y_{<i}, \mathcal{T})$, where $y$ is the visual token of an image with $S$ as the sequence length, $\mathcal{T}$ is the text condition.

**Reward Calculation.** The overall design philosophy of our reward model is to leverage QA-based visual comprehension models (Chen et al., 2024; Liu et al., 2024; Bai et al., 2025b). Specifically, we employ InternVL3.5-38B (Wang et al., 2025b) to provide appropriate incentives, returning a consistency score $\text{R}_\mathcal{I} \in [0, 1]$ for each text-image pair. For each text-image pair, we query the reward model with multiple "yes or no" questions regarding semantic alignment and minute-detail consistency. The global question evaluates text-image semantic overall alignment using the prompt: "*Does this image match the description? Please directly respond with yes or no*". Subsequently, we decompose the prompt into semantic tuples (*e.g.*, attributes and spatial relations) and generate yes-or-no sub-questions, including specific inquiries addressing the fine-grained differences between paired prompts, to evaluate whether the image corresponding to the first prompt within a group aligns solely with the first prompt—and not the second—regarding the

differing fine-grained semantics.

We record the probability of the model responding with "Yes" as $P_{yes}$ and "No" as $P_{no}$, calculating the reward score as $S(\mathcal{I}, \mathcal{P}) = P_{yes}/(P_{yes} + P_{no})$. In summary, the reward computation essentially entails the MLLMs performing a VQA task for the prompt and generated image to return a score between 0 and 1 for each question, with the final reward obtained by averaging the MLLM evaluations across multiple questions for a prompt.

**Reinforcement Learning.** Our proposed Pair-GRPO is an improved version of GRPO, with training prompts sourced from `FocusDiff-Data`. We set the $G = 12$, first expanding the group size from 12 to 24. There is a probability $p$ that the group size may further increase to 28, as we introduce ground-truth images corresponding to prompt pairs from `FocusDiff-Data` and pair them with the prompts. The probability $p$ is dynamic, decreasing from 1.0 at the start of training to 0.0 by the end.

During RL training, we employ the fine-tuned X-Omni as the backbone model. We freeze the tokenizer (including a SigLIP encoder and a diffusion decoder) while keeping all other parameters tunable. We conduct a total of 3,000 iterations of post-training optimization. To enhance training stability, we identify the learning rate as a crucial hyperparameter: an excessively small learning rate results in insignificant performance gains, while an overly large one leads to unstable training. To address this, we design a combined Linear-Cosine learning rate scheduler. The learning rate linearly drops from a peak value to a lower "convert learning rate" at a "convert step", and then gradually decreases along a cosine curve. However, we still encounter some instability during training, indicated by a downward trend in the reward curve. To address this, we adopt the following measures:

(1) when the reward curve drops sharply, we reduce the learning rate to half or two-thirds of its current value and resume training;

(2) when the reward curve declines gradually, implying that the KL divergence constraint with a less capable reference model limits model improvement, we update the reference model to the current model and resume training.

## F. Evaluation Details

**Baseline.** We compare X-Omni-FocusDiff against two primary categories of models: diffusion methods, and AR-based methods. The diffusion baselines include SD3 (Esser et al., 2024), FLUX.1-dev (Labs, 2024), Sana-1.5 (Xie et al., 2025a), Flow-GRPO (Liu et al., 2025), Lumina-Image 2.0 (Qin et al., 2025), HiDream-I1-Full (Cai et al., 2025b), HunyuanImage-2.1 (Team, 2025), Qwen-Image (Wu et al.,

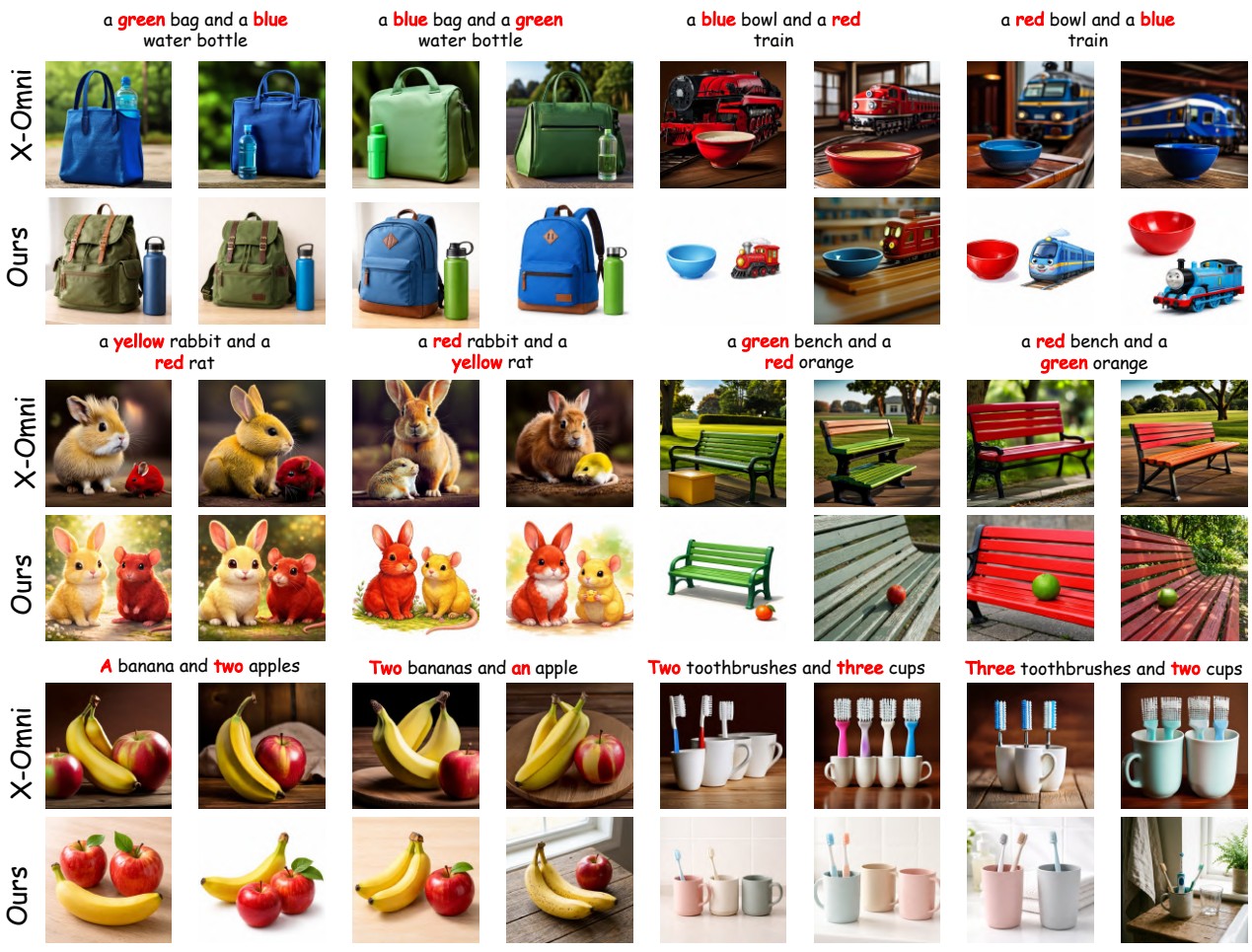

*Figure 9.* More qualitative Comparisons between X-Omni-FocusDiff and X-Omni on similar prompts.

2025b), Z-Image (Cai et al., 2025a), and FLUX.2-dev (Labs, 2025). Notably, FLUX.2 Dev is a 32B large DiT (Peebles & Xie, 2023) model with another 24B model (AI, 2025) for text encoding. And Qwen-Image is a 20B DiT model with another 7B model (Bai et al., 2025b) for text encoding.

The AR-based baselines include SEED-X (Ge et al., 2024), Show-o2 (Xie et al., 2025b), Emu3 (Wang et al., 2024), BLIP-3o (Chen et al., 2025a), OmniGen2 (Wu et al., 2025c), Bagel-Think (Deng et al., 2025), Janus-Pro (Chen et al., 2025b), Lumina-mGPT 2.0 (Xin et al., 2025), T2I-R1 (Jiang et al., 2025), X-Omni (Geng et al., 2025), GLM-Image (Team, 2026b). Among them, Emu3, Lumina-mGPT 2.0, Janus-Pro, and T2I-R1 operate as purely AR methods where discrete tokens are derived via VQVAE (Esser et al., 2021) encoding without the integration of diffusion models; in contrast, SEED-X, Show-o2, BLIP-3o, Omni-Gen2, Bagel-Think, X-Omni, and GLM-Image combine AR and diffusion paradigms, utilizing autoregressive inference to generate the requisite discrete tokens which are subsequently rendered into images by a diffusion model.

**DeltaBench.** In DeltaBench, we leverage Qwen3-VL-30B (Bai et al., 2025a) as the primary evaluation model. For each image-prompt pair, we query the VLM with binary (Yes/No) questions to assess both global alignment and minute-detail consistency. The first binary questions evaluates global semantics using the prompt: *"Does this image match the description? Please directly respond with yes or no."* Subsequently, we employ two specific questions referring to the fine-grained differences between paired prompts, ensuring that the image generated from $\mathcal{T}^1$ aligns strictly with $\mathcal{T}^1$ rather than $\mathcal{T}^2$. For instance, given a scenario where $\mathcal{T}^1 =$ *"A professional chef in a bright, bustling kitchen plating a gourmet steak, with a crack on the white tiles in the far upper-left corner of the backsplash"* and $\mathcal{T}^2 =$ *"A professional chef in a bright, bustling kitchen plating a gourmet steak, with perfectly smooth white tiles in the far upper-left corner of the backsplash"*, to evaluate the semantic alignment of the images generated from $\mathcal{T}^1$, one specific question is *"Is there a crack on the white tiles in the far upper-left corner of the backsplash? Please directly respond with yes or no. If the mentioned attribute is irrelevant to the image content, please answer no"* while

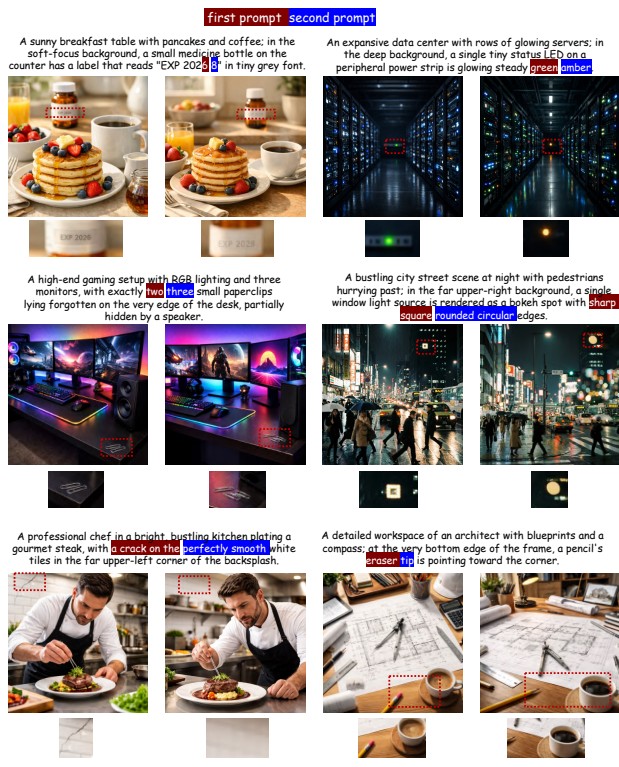

*Figure 10.* Qualitative examples of X-Omni-FocusDiff on similar prompts in DeltaBench.

another specific question is *"Are the white tiles in the far upper-left corner of the backsplash not perfectly smooth? Please directly respond with yes or no. If the mentioned attribute is irrelevant to the image content, please answer no"*. We denote the binary question evaluating global semantics as $Q_1$ and the two specific questions as $Q_2$ and $Q_3$. For each question $Q_k$, we record the probability of the model responding "yes" (denoted $P_{\text{yes}}^{Q_k}$) and "no" (denoted $P_{\text{no}}^{Q_k}$), calculating the semantic consistency score as

$$S(\mathcal{I}, \mathcal{T}) = \frac{1}{4} \left[ \frac{2P_{\text{yes}}^{Q_1}}{P_{\text{yes}}^{Q_1} + P_{\text{no}}^{Q_1}} + \frac{P_{\text{yes}}^{Q_2}}{P_{\text{yes}}^{Q_2} + P_{\text{no}}^{Q_2}} + \frac{P_{\text{yes}}^{Q_3}}{P_{\text{yes}}^{Q_3} + P_{\text{no}}^{Q_3}} \right].$$

Following this evaluation strategy, we require the text-to-image model to generate two images for each prompt. Therefore, for a pair of similar prompts $(\mathcal{T}_i^1, \mathcal{T}_i^2)$, we obtain four generated images $(\mathcal{I}_i^{1,1}, \mathcal{T}_i^{1,2}, \mathcal{I}_i^{2,1}, \mathcal{I}_i^{2,2})$. We then compute the semantic consistency scores for each image with respect to its corresponding prompt: $s_i^{1,1} = S(\mathcal{I}_i^{1,1}, \mathcal{T}_i^1)$, $s_i^{1,2} = S(\mathcal{I}_i^{1,2}, \mathcal{T}_i^1)$, $s_i^{2,1} = S(\mathcal{I}_i^{2,1}, \mathcal{T}_i^2)$, $s_i^{2,2} = S(\mathcal{I}_i^{2,2}, \mathcal{T}_i^2)$. The arithmetic mean score is calculated as: $s_a = \frac{1}{4N} \sum_{i=1}^{N}(s_i^{1,1} + s_i^{1,2} + s_i^{2,1} + s_i^{2,2})$, and the geometric mean score is calculated as: $s_g = \frac{1}{N} \sqrt[4]{s_i^{1,1} \cdot s_i^{1,2} \cdot s_i^{2,1} \cdot s_i^{2,2}}$. The score of the geometric (arithmetic) mean for "Average" is obtained by averaging the geometric (arithmetic) mean scores of the six sub-tasks.

**Existing Benchmarks.** Furthermore, we also conduct zero-shot evaluation on 3 existing text-to-image benchmarks: GenEval (Ghosh et al., 2023), LongText-Bench (Geng et al., 2025), and DPG-Bench (Hu et al., 2024). GenEval contains 6 different subtasks of varying difficulty requiring various compositional skills, including `single object` (Single Obj.), `two objects` (Two Obj.), `counting`, `colors`, `position`, `color binding` (Color Attr.). And we adopt the metric proposed by (Ghosh et al., 2023) for evaluation. Each subtask is scored independently, and the overall score is calculated as the average of all six subtask scores. The LongText-Bench is a benchmark designed to evaluate the capability of text-to-image models in accurately rendering long and complex text sequences within generated images. While for DPG-Bench, we follow the metrics proposed in (Hu et al., 2024) to conduct evaluation.

## G. More Experimental Results

Figures 9 presents a qualitative comparison between X-Omni-FocusDiff and X-Omni using similar prompts that differ only in fine-grained semantics. It is evident that X-Omni struggles to precisely adhere to fine-grained constraints, often generating images that deviate significantly from the prompt requirements. In contrast, our X-Omni-FocusDiff consistently generates high-quality images that faithfully meet these specifications.

Figure 10 presents detailed qualitative examples of X-Omni-FocusDiff on DeltaBench, highlighting a key strength of our model: beyond aligning with the primary subject semantics, it demonstrates precise control over intricate, off-center details. Whether handling the specific geometry of background bokeh, text on a peripheral bottle label, or the color of minute distant lights, our model satisfies these fine-grained constraints with remarkable accuracy. Notably, these target elements are situated in inconspicuous locations—such as backgrounds or corners—or are extremely small in scale, distinct from the main objects in the image. The rigorous mastery of such subtle features provides intuitive evidence of our method's exceptional fine-grained controllability, ensuring the generated images are strictly faithful to the prompt requirements. In the future, we aim to extend our approach through further exploration to elevate image generation models to the next level of intelligence (Wu et al., 2026; Niu et al., 2025; Qiu et al., 2026; Pan et al., 2025a). Furthermore, we plan to apply this method to broader scenarios, including video generation (Wu et al., 2025a; Pan et al., 2026a; Lin et al., 2025; Wan et al., 2025) and and other diverse applications (Zhang et al., 2026; Bu et al., 2025; Jiang et al., 2026).

