# OpenReview forum: "Benchmarking and Improving Fine-Grained Text-to-Image Alignment via Paired Reinforcement Learning"
_ICML.cc/2026/Conference — ICML 2026 regular_

### Official Review · Reviewer_M7Nm · 2026-03-01

**Soundness:** 4
**Presentation:** 3
**Significance:** 4
**Originality:** 3
**Overall Recommendation:** 5
**Confidence:** 5

**Summary:**

This paper studies fine-grained prompt adherence in text-to-image generation and shows AR models often miss subtle word-level differences. It introduces DeltaBench, a paired-prompt benchmark that measures both alignment and stability. To improve this, it proposes FineFocus (paired training data + Pair-GRPO), and reports large gains on DeltaBench and other standard benchmarks.

**Compliance With Llm Reviewing Policy:**

Affirmed.

**Final Justification:**

The authors have satisfactorily addressed my concerns. I will maintain my positive score and recommend acceptance.

**Key Questions For Authors:**

Please see weaknesses

**Limitations:**

yes

**Strengths And Weaknesses:**

**Strengths:**

1. This paper has a clear motivation, and the proposed DeltaBench provides a rigorous test of T2I models’ fine-grained understanding and controllability.

2. Collecting the FineFocus dataset and proposing the paired-GRPO optimization are solid contributions. The authors’ experimental pipeline is complete and well-structured.

3. The benchmark uses open-source evaluation models, which improves reproducibility.

**Weaknesses:**

I only have a few concerns and suggestions.

1. For Nano-Banana, the paper only provides a few qualitative examples. I suggest adding quantitative experiments on it to broaden the benchmark’s coverage and strengthen the reliability of the proposed model.

2. The authors only validate the effectiveness of FineFocus-Data and Pair-GRPO on the autoregressive model X-Omni. I suggest also adding, in Table 1, experiments that fine-tune diffusion-based models using FineFocus-Data, to demonstrate broader applicability.

3. The authors could add a discussion of the limitations and failure cases of FineFocus.

---

> ### Author Rebuttal · Authors · 2026-03-31
>
> We sincerely thank you for the professional comments! We are encouraged that FineFocus and pair-GRPO are recognized as solid contributions. We will explain your concerns as follows.
>
> >### W1: Quantitative experiments for Nano Banana.
>
> **A1:** Thank you for the suggestion. We evaluate Nano Banana Pro on DeltaBench, with the results shown in the table below. As a widely recognized, leading proprietary model for image generation, Nano Banana Pro achieves SoTA average scores on DeltaBench, outperforming other academic models, including X-Omni-FineFocus. ***This is entirely reasonable, as industrial-grade models benefit from significantly more computational resources and training data than academic models like ours.***
>
> However, we also find several important observations.
>
> **(1)** Even for Nano Banana Pro, the performance of $S_a$ is still noticeably higher than $S_g$, resulting in a averahe performance gap ($S_a-S_g$) larger than that of X-Omni-FineFocus. This suggests that even industrial-scale models struggle with stability when generating fine-grained local details. ***In contrast, the smaller $S_a-S_g$ gap of X-Omni-FineFocus demonstrates its improved generation stability.***
>
> **(2)** On counting subtask, our X-Omni-FineFocus achieves higher $S_a$ and $S_g$ scores than Nano-Banana Pro, ***indicating stronger performance on counting-related cases.***
>
> **(3)** We also identify many qualitative cases within DeltaBench (Fig. 4 of our paper) where X-Omni-FineFocus generates images that are better aligned with the prompts than those generated by Nano-Banana Pro.
>
> Given that Nano-Banana Pro is widely regarded as the strongest product-level model, these results further underscore the robust capabilities of our model.
>
> **Table1:** DeltaBench results for Nano Banana Pro.
> Models|Appear-$S_a$|Appear-$S_g$|Color-$S_a$|Color-$S_g$|Count-$S_a$|Count-$S_g$| Pos.-$S_a$|Pos.-$S_g$|Style-$S_a$|Style-$S_g$|Text-$S_a$|Text-$S_g$|**Avg-$S_a$**|**Avg-$S_g$**
> |-|-|-|-|-|-|-|-|-|-|-|-|-|-|-|
> |Nano Banana Pro|77.9|73.0|95.0|92.5|59.9|54.4|73.9|60.7|95.5|93.0|98.7|96.2|83.5|78.3
>
> &nbsp;
>
> >### W2: Adding experiments on more backbone models using FineFocus-Data
>
> **A2:** Thank you for the suggestion. Our current backbone X-Omni is an auto-regressive model with a diffusion decoder. To demonstrate the broad applicability of our method, we further select Sana-1.5 (a diffusion-based model) and Janus-Pro (a pure autoregressive model) as alternative backbones, developing Sana-FineFocus and Janus-FineFocus in a similar way to X-Omni-FineFocus. As shown in the table below, **both Janus-FineFocus and Sana-FineFocus consistently and significantly outperform their corresponding backbone models on all DeltaBench subtasks.** This demonstrates that **our method is model-agnostic and applies well across different backbone paradigms**, including diffusion-based, pure AR-based, and hybrid AR-diffusion architectures.
>
> **Table2:** DeltaBench results for Sane-FineFocus and Janus-FineFocus.
> Models|Appear-$S_a$|Appear-$S_g$|Color-$S_a$|Color-$S_g$|Count-$S_a$|Count-$S_g$| Pos.-$S_a$|Pos.-$S_g$|Style-$S_a$|Style-$S_g$|Text-$S_a$|Text-$S_g$|**Avg-$S_a$**|**Avg-$S_g$**
> -|-|-|-|-|-|-|-|-|-|-|-|-|-|-
> Sana-1.5|68.2|60.2|90.8|88.6|40.4|33.2|43.0|32.1|87.5|83.4|61.2|54.0|65.2|58.6
> Sana-FineFocus|69.0|64.8|91.0|89.0|56.5|50.8|56.0|42.5|88.5|84.6|85.0|82.0|74.3|69.0
> Janus-Pro|65.4|56.6|86.8|84.5|25.5|17.0|44.2|25.9|85.3|81.0|49.1|37.9|59.4|50.6
> Janus-FineFocus|68.5|63.5|89.5|86.5|52.0|45.0|54.0|40.0|87.5|83.0|78.5|74.5|71.7|65.4
>
> &nbsp;
>
> >### W3: Adding a discussion of the limitations and failure cases of FineFocus.
>
> **A3:** We apologize for omitting the discussion on limitations and failure cases in our current manuscript. We summarize two main limitations of X-Omni-FineFocus below.
>
> (1) Our FineFocus-Data is constructed from the open-source GPT-Image-Edit-1.5M and Nano-banana-150K datasets. While it encompasses numerous high-quality before-and-after editing pairs and covers a diverse range of editing types, certain fine-grained visual differences, such as depth-related perspective variations, remain underrepresented. Consequently, this limitation can result in unstable 3D-perspective-based object manipulation during the image generation process. In future work, we will continue to expand the overall scale of FineFocus-Data.
>
> (2) Our main model leverages X-Omni as the backbone. While X-Omni possesses basic text-rendering capabilities, it struggles to generate text-rich images with complex layouts, such as posters. Constrained by these inherent backbone limitations, X-Omni-FineFocus consequently exhibits similar shortcomings in these complex text-heavy generation scenarios. In future work, we aim to enhance the model's capacity to synthesize rich-text images within intricate layouts.
>
> In the next version of the paper, we will include a detailed discussion of these limitations, accompanied by visual examples of failure cases.

---

> > ### Author Rebuttal · Reviewer_M7Nm · 2026-03-31
> >
> > Thank you for the rebuttal. The authors have satisfactorily addressed my concerns. I will maintain my positive score and recommend acceptance. Good luck!

---

> > > ### Author Response · Authors · 2026-04-01
> > >
> > > We are grateful for your professional suggestions, which have significantly contributed to improving our paper! We will integrate the refined analyses and the new experimental results into the next version of our manuscript.

---

### Official Review · Reviewer_RVYF · 2026-03-07

**Soundness:** 2
**Presentation:** 3
**Significance:** 3
**Originality:** 2
**Overall Recommendation:** 4
**Confidence:** 3

**Summary:**

This paper addresses the poor fine-grained semantic alignment ability of text-to-image models, proposes the FineFocus framework, and constructs the DeltaBench benchmark to evaluate the fine-grained control capability of models.It builds the paired dataset FineFocus-Data based on image editing data and designs the Pair-GRPO reinforcement learning algorithm, enabling the model to learn subtle semantic differences from similar text-image pairs. Extensive experiments show that FineFocus significantly improves the fine-grained alignment accuracy and generation stability of T2I models, outperforming existing methods on multiple benchmarks.

**Compliance With Llm Reviewing Policy:**

Affirmed.

**Final Justification:**

My concerns have been solved and I will maintain my rating.

**Key Questions For Authors:**

see weaknesses

**Strengths And Weaknesses:**

**Strengths**

1. The paper provides thorough empirical validation including ablation studies, evaluations on DeltaBench and three standard benchmarks, and human-correlation analysis for VLM-based metrics.

2. An important area discussed by the article is fine-grained semantic alignment in text-to-image generation, a key limitation for real-world use cases such as design and content creation.

3. The work introduces a new perspective that enhances alignment by learning subtle differences from similar text–image pairs. The combination of pairwise editing data (FineFocus-Data) and adapted RL (Pair-GRPO) is innovative and distinct from conventional global alignment methods, without breaking the original autoregressive training pipeline.

**Weaknesses**

1. The paper lacks ablation on critical hyperparameters (e.g., group size G, annealing probability p) and only validates the method on the X-Omni backbone. No experiments on other T2I models are provided, weakening claims about universality.

2. FineFocus also lacks evaluation on extremely long-tailed characters or complex multi-text scenes, restricting its real-world impact and applicability.

3. Pair-GRPO is a straightforward extension of vanilla GRPO (paired group expansion + GT guidance) with no novel theoretical contributions or new RL foundations. The method relies heavily on existing techniques, leading to limited originality.

4. The fine-grained T2I alignment is an incremental problem comparing with the general T2I alignment. I would doubt whether this problem is as that important or can we solve it by prior alignment methods.

---

> ### Author Rebuttal · Authors · 2026-03-31
>
> Thank you for the valuable comments! We will explain your concerns as follows.
>
> >### W1: Lacking ablation on hyperparameters and no experiments on other T2I models
>
> **A1: (1.1)** We apologize for omitting our ablation studies on $G$ and $p$ in the manuscript. As shown below, our evaluation of $G \in \{4, 8, 12\}$ indicates that $G=12$ yields the highest average performance on DeltaBench. Regarding $p$, comparing constant values of $\{1.0, 0.0, 0.5\}$ against a dynamic value ($1.0 → 0.0$) reveals that the dynamic setup achieves the best results. Thus, we adopt these optimal values as our final configuration.
>
> **Table1:** Average performance on DeltaBench with different $G$ or $p$
> $G$|Avg-$S_a$|Avg-$S_g$
> -|-|-
> 4|73.2|67.2
> 8|75.3|71.4
> 12|**77.1**|**72.8**
> **$p$**|**Avg-$S_a$**|**Avg-$S_g$**
> 0.0|76.1|71.0
> 0.5|76.7|72.0
> 1.0|75.9|71.2
> 1.0→0.0|**77.1**|**72.8**
>
> **(1.2)** To demonstrate the universality of our method, we also develop Sana-FineFocus and Janus-FineFocus using Sana-1.5 (diffusion-based) and Janus-Pro (AR-based) as backbones. As shown below, both variants consistently outperform their respective baselines across all DeltaBench subtasks. This confirms that **our approach is model-agnostic and generalizes effectively across different architectures.**
>
> **Table2:** DeltaBench results for Sane-FineFocus and Janus-FineFocus.
> Models|Appear-$S_a$|Appear-$S_g$|Color-$S_a$|Color-$S_g$|Count-$S_a$|Count-$S_g$| Pos.-$S_a$|Pos.-$S_g$|Style-$S_a$|Style-$S_g$|Text-$S_a$|Text-$S_g$|**Avg-$S_a$**|**Avg-$S_g$**
> -|-|-|-|-|-|-|-|-|-|-|-|-|-|-
> Sana-1.5|68.2|60.2|90.8|88.6|40.4|33.2|43.0|32.1|87.5|83.4|61.2|54.0|65.2|58.6
> Sana-FineFocus|69.0|64.8|91.0|89.0|56.5|50.8|56.0|42.5|88.5|84.6|85.0|82.0|74.3|69.0
> Janus-Pro|65.4|56.6|86.8|84.5|25.5|17.0|44.2|25.9|85.3|81.0|49.1|37.9|59.4|50.6
> Janus-FineFocus|68.5|63.5|89.5|86.5|52.0|45.0|54.0|40.0|87.5|83.0|78.5|74.5|71.7|65.4
>
> >### W2: Lacking evaluation on extremely long-tailed characters or complex multi-text scenes
>
> **A2: Regarding complex multi-text scenes**, we have already included an evaluation on LongText-Bench (Table 2 of Paper), which shows X-Omni-FineFocus improves performance from 0.900 to 0.935 over the X-Omni backbone.
>
> **Regarding extremely long-tailed characters**, we construct a dedicated test set and conduct a human evaluation (rated 1 to 5) comparing X-Omni and X-Omni-FineFocus. As shown below, X-Omni-FineFocus achieves superior performance over the X-Omni backbone and also outperforms GLM-Image. These validate **the enhanced capability of X-Omni-FineFocus in handling both extremely long-tailed characters and complex multi-text scenes.**
>
> **Table3:** Performance on extremely long-tailed characters
> Models|Score↑
> -|-
> X-Omni|2.3
> GLM-Image|2.9
> **X-Omni-FineFocus**|**3.2**
>
> >### W3: Pair-GRPO is a straightforward extension of vanilla GRPO with no new RL foundations
>
> **A3:** To clarify, our primary contribution is not proposing a fundamentally new RL theory. Instead, we aim to develop a suitable algorithmic refinement tailored to better align with our core motivations and design principles.
>
> **The central novelty of our paper lies in its high-level design paradigm**: shifting from global vision-language alignment using single text-image pairs to learning the subtle differences between similar pairs for fine-grained alignment. This conceptual shift naturally drives our most significant contributions—the innovative data construction pipeline and the DeltaBench benchmark, as acknowledged by the other two Reviewers. Consequently, Pair-GRPO should be viewed as a specialized RL adaptation for fine-grained T2I alignment rather than a standalone theoretical foundation, and our empirical results demonstrate its overall effectiveness.
>
> >### W4: Fine-grained T2I alignment is incremental comparing with general T2I alignment
>
> **A4:** While fine-grained alignment might have appeared incremental a year ago when early T2I models still struggled with global alignment, current models already excel in general alignment. This rapid progress has exposed **fine-grained alignment as a primary failure mode**, where localized off-center prompt details are frequently overlooked. Furthermore, **fine-grained alignment is crucial in practice**, as the omission of even a subtle, user-defined attribute can render a general coherent image unusable for real-world applications. Therefore, fine-grained alignment demands substantial attention, necessitating dedicated methods and benchmarks to drive the next stage of T2I progress.
>
> Furthermore, fine-grained T2I alignment cannot be solved by prior methods, which typically optimize global semantic consistency with independent text-image pairs. While EvoGen also constructs contrastive training pairs, it mainly permutes core attributes to improve compositional learning rather than targeting overlooked fine-grained details. In contrast, our method is designed specifically to strengthen sensitivity to such minor yet crucial details beyond the core semantics.

---

> > ### Author Rebuttal · Reviewer_RVYF · 2026-04-03
> >
> > Overall Recommendation:

---

> > > ### Author Response · Authors · 2026-04-03
> > >
> > > Thank you again for your precious time and valuable suggestions.

---

### Official Review · Reviewer_raz1 · 2026-03-12

**Soundness:** 2
**Presentation:** 3
**Significance:** 3
**Originality:** 2
**Overall Recommendation:** 4
**Confidence:** 5

**Summary:**

This paper studies fine-grained text-to-image alignment for autoregressive image generation models, arguing that current AR systems often miss subtle prompt differences even when they handle global semantics reasonably well. To evaluate this, the authors introduce DeltaBench, a paired-prompt benchmark where each example consists of two highly similar prompts differing in a small but visually meaningful detail. To improve performance, they propose FineFocus, which combines a paired training dataset, FineFocus-Data, with a modified GRPO-style reinforcement learning procedure called Pair-GRPO. Experiments built on X-Omni show substantial gains on DeltaBench and smaller but generally positive gains on several existing text-to-image benchmarks.

**Compliance With Llm Reviewing Policy:**

Affirmed.

**Final Justification:**

After reading all the authors’ and other reviewers’ comments, I decide to raise my score to 4.

**Key Questions For Authors:**

1. How much improvement comes from Pair-GRPO versus the paired dataset itself? Could the authors provide a strong SFT baseline trained on FineFocus-Data without RL?
2. Pair-GRPO normalizes rewards across samples from different prompts. How do the authors ensure that reward distributions across prompts are comparable and do not introduce instability?
3. Does the method generalize to other AR-based T2I models beyond X-Omni, or is the improvement largely backbone-dependent?

**Limitations:**

yes

**Strengths And Weaknesses:**

### Strengths
- In general this paper is well-written and easy to follow.
- The data construction idea is thoughtful. Using image editing pairs to obtain near-matched images with localized differences is a clever way to reduce nuisance variation.
- The empirical gains on the proposed benchmark are substantial relative to the X-Omni backbone. In Table 1, the jump from 61.8 to 77.1 in average ($s_a$), and from 53.2 to 72.8 in average ($s_g$), is large enough that this is not just noise-level improvement.

### Weaknesses
- DeltaBench relies heavily on VLM-based automatic evaluation rather than human annotations. Although the paper reports correlation with human judgments, the benchmark still depends on a single evaluation model for most experiments. This raises concerns about evaluation bias or reward-model overfitting, especially since similar VLMs are used during training and evaluation.
- Pair-GRPO mixes samples from different prompts within the same group for advantage normalization. Since reward distributions may differ across prompts, this cross-prompt normalization may introduce noisy policy updates and lacks clear theoretical justification.
- The proposed framework relies on a large paired dataset derived from image editing pipelines. While effective, this setup requires specialized data collection and filtering procedures, which may limit general applicability. It remains unclear whether the method would still be effective when paired data is unavailable or constructed differently.

---

> ### Author Rebuttal · Authors · 2026-03-31
>
> Thank you for the valuable comments! We respond to your concerns as follows.
>
> >### (1) Weak1: Reliance on VLM-based automatic evaluation
>
> **A1:**
> **1.1) Necessity of VLM-based Evaluation.** Human evaluation is accurate but costly, subjective, and difficult to scale. VLM-based evaluation has become a standard practice in recent T2I benchmarks, e.g., WISE.
>
> **1.2) Mitigating Evaluation Bias and Reward-model Overfitting.** We strictly decouple our training-time reward model (InternVL3.5-38B) from DeltaBench evaluator (Qwen3-VL-30B). We choose Qwen3-VL-30B for evaluation as it shows strongest consistency with human rating among open-source models (Table 4 of paper).
>
> **1.3) Additional Evaluations.** To further reduce single-evaluator bias, we use two advanced closed-source VLMs (GPT-5, Gemini3-Pro) as evaluators. We ask them to directly assign a score from 0 to 10 instead of relying on the probability of "yes" response for scoring. As shown below, **X-Omni-FineFocus consistently outperforms its backbone, as well as GLM-Image and T2I-R1 under both GPT-5 and Gemini3-Pro**, further supporting its effectiveness. Besides, the performance ranking across these four models remains unchanged compared to Paper-Table1, suggesting our original evaluation was objective.
>
> **Table1:** AVG DeltaBench results with GPT-5(GPT)↑ and Gemini3-Pro(Gem)↑.
> Model|Avg-$S_a$-GPT|Avg-$S_g$-GPT|Avg-$S_a$-Gem|Avg-$S_g$-Gem|
> -|-|-|-|-
> T2I-R1|5.8|4.8|6.3|5.5
> X-Omni|5.2|4.1|5.9|4.7
> GLM-Image|6.5|5.6|7.0|6.3
> X-Omni-FineFocus|**7.1**|**6.3**|**7.4**|**6.8**
>
> >### (2) Weak2 & Q2: Pair-GRPO mixes samples from different prompts for ADV normalization
>
> **A2:** To clarify, the prompt pairs used in Pair-GRPO are NOT arbitrary texts with disparate reward characteristics. Indeed, each pair is derived from nearly identical images and generated by a VLM into the same template, typically differing by only 1–3 words on non-primary attributes (e.g.,background color). Thus, **these semantically aligned prompt pairs have comparable reward distributions and correlated reward expectations**. And cross-prompt normalization does NOT introduce noisy updates; instead, it encourages sharper discrimination of subtle semantic differences.
>
> Our empirical results support this claim: **(1)** Fig.5(c) of paper shows that Pair-GRPO is more stable and converges faster than vanilla GRPO. **(2)** As shown below, **Pair-GRPO trained on FineFocus-Data outperforms vanilla GRPO on DeltaBench, while applying Pair-GRPO to unrelated prompt pairs leads to noisy updates and worse optimization.** This confirms cross-prompt normalization is effective with our semantically matched pairs.
>
> **Table2:** AVG DeltaBench results.
> ||Avg-$S_a$|Avg-$S_g$
> -|-|-
> GRPO+FineFocus-Data|71.9|66.0
> Pair-GRPO+FineFocus-Data|77.1|72.8
> Pair-GRPO+Unrelated-Pairs|67.6|60.7
>
> >### (3) Weak3: Whether the method is effective when paired data is unavailable
>
> **A3:** First, we clarify that **near-matched image-editing pairs are a crucial ingredient for Pair-GRPO, and acquiring them is cost-effective**. Many general-purpose image editing datasets are publicly available and can be adapted into FineFocus-Data via lightweight VLM filtering. Even without such datasets, open-source models like Qwen-Image-Edit can generate the needed image pairs, making data collection accessible.
>
> Then, we explore the setting with paired editing data completely unavailable. By generating only semantically related prompt pairs with LLMs, we introduce an ablated Pair-GRPO variant that does not inject any ground-truth image during RL(also skips SFT). As shown below, this variant(w/o Pair images) still outperforms X-Omni and X-Omni+GRPO, while underperforming X-Omni-FineFocus. This suggests **our method remains effective without paired images, while paired images unlock larger gains**.
>
> **Table4:** AVG DeltaBench results
> Model|Avg-$S_a$|Avg-$S_g$
> -|-|-
> X-Omni|61.8|53.2
> X-Omni+GRPO|71.9|66.0
> X-Omni-FineFocus|77.1|72.8
> w/o pair-images|73.3|68.1
>
> >### (4) Q1: How much improvement comes from Pair-GRPO vs the paired dataset itself
>
> **A4:** Table below reports DeltaBench results of X-Omni after SFT on FineFocus-Data w/o RL (X-Omni-SFT). **SFT alone brings only marginal gains, while Pair-GRPO contributes the majority of improvement**. Thus, FineFocus-Data alone is insufficient and the best performance comes from combining FineFocus-Data with Pair-GRPO.
>
> **Table3:** AVG DeltaBench results
> Model|Avg-$S_a$|Avg-$S_g$
> -|-|-
> X-Omni|61.8|53.2
> X-Omni-SFT|66.9|59.9
> X-Omni-FineFocus|77.1|72.8
>
> >### (5) Q3: Does the method generalize to other models
>
> **A5:** We further apply Janus-Pro as backbone to develop Janus-FineFocus. As shown, **Janus-FineFocus significantly outperforms Janus-Pro on DeltaBench**, confirming that our method generalizes to other AR-based T2I models. Detailed results of Janus-FineFocus are provided in Table2 of Rebuttal to M7Nm.
>
> **Table5:** AVG DeltaBench results
> Model|Avg-$S_a$|Avg-$S_g$
> -|-|-
> Janus-Pro|59.4|50.6
> Janus-FineFocus|71.7|65.4

---

> > ### Author Rebuttal · Reviewer_raz1 · 2026-04-03
> >
> > Thank you for your rebuttal. After reading all the authors’ and other reviewers’ comments, I decide to raise my score to 4.

---

> > > ### Author Response · Authors · 2026-04-03
> > >
> > > Thank you for raising the score! We will integrate the new experimental results into the next version of our manuscript.

---

### Decision · Program_Chairs · 2026-04-30

**Decision:**

Accept (regular)

**Comment:**

The authors present FineFocus, which addresses an important problem in fine-grained text-to-image alignment by introducing DeltaBench, a paired-prompt benchmark that rigorously evaluates subtle semantic differences, and proposing the FineFocus framework combining paired training data with Pair-GRPO reinforcement learning. Reviewers appreciated the thoughtful data construction using image editing pairs, the substantial empirical gains on DeltaBench (average scores improving from 61.8 to 77.1 for alignment and 53.2 to 72.8 for stability), the comprehensive experimental validation, and the commitment to open-source release. Initial concerns about VLM-based evaluation bias, cross-prompt normalization in Pair-GRPO lacking theoretical justification, limited backbone validation beyond X-Omni, scalability of paired dataset requirements, and missing evaluations on long-tailed characters were thoroughly addressed in rebuttal through additional experiments with GPT-5 and Gemini3-Pro evaluators, validation on Sana-1.5 and Janus-Pro backbones, hyperparameter ablations, SFT baseline comparisons, and targeted evaluations on challenging cases. All three reviewers confirmed their concerns were fully resolved, with two maintaining scores of 4 (weak accept) and one maintaining a score of 5 (accept). While the paper has limitations including modest algorithmic novelty as primarily an extension of GRPO and reliance on specialized paired datasets, the work makes a solid contribution by establishing a principled evaluation framework and demonstrating effective methods for improving fine-grained alignment. AC feel the work is likely to benefit the text-to-image generation community, thus recommends accept.